# Bacteria encode post-mortem protein catabolism that enables altruistic nutrient recycling

Savannah E. R. Gibson[1], Isabella Frost[1], Stephen J. Hierons[1], Tessa Moses [2], Wilson C. K. Poon[3], Stuart A. West [4] & Martin J. Cann [1,5] ✉

Bacterial death is critical in nutrient recycling. However, the underlying mechanisms that permit macromolecule recycling after bacterial death are largely unknown. We demonstrate that bacteria encode post-mortem protein catabolism via Lon protease released from the dead bacteria. Growth assays reveal that the lysate of Lon protease-null bacteria does not provide a growth benefit to wild type cells. This deficiency is reversed with exogenous recombinant Lon protease, confirming its post-mortem role and is independent of Lon ATPase activity. Biochemistry, growth assays and metabolomics demonstrate that Lon protease facilitates peptide nutrient release, benefitting living cells and acting as a cooperative public good. We also show that the production of Lon protease cannot be explained by a personal benefit to living cells. Although Lon protease can also provide a benefit to living cells under stressful conditions by helping control protein quality, this private benefit does not outweigh the cost under the conditions examined. These results suggest that Lon protease represents a post-mortem adaptation that can potentially be explained by considering the post-mortem indirect benefit to other cells (kin selection). This discovery highlights an unexpected post-mortem biochemistry, reshaping our understanding of nutrient recycling.

Most bacteria exhibit little growth in their natural environments, with nutrient limitation being a typical causative factor[1]. Therefore, how bacteria catabolise environmental macromolecules and import the resulting nutrients is of general importance for a broader understanding of microbial growth[2,3]. Bacteria typically transport small peptides into the cell to serve as a source of fixed nitrogen and amino acids[4]. This transport is exploited in laboratory microbial culture when casein (the predominant protein component of milk) degraded with enzymes (a casein hydrolysate) is used as a source of nutrients for growth. Prior degradation is necessary because whole proteins are typically too large for bacterial uptake, enzyme degradation produces amino acids and small peptides that can be transported into the bacterium to be used for growth, for example in the almost universally used LB broth culture medium[5]. Casein hydrolysate gives a dose-dependent enhancement of wild type *Escherichia coli* growth when used as an additive to minimal media with glycerol as a carbon source (Fig. 1a). In line with this observation, one may expect that the proteins released from the lysis of dead bacteria in the environment must be catabolised to small peptides before they can support the growth of other live cells. Here, we show that this is indeed the case and demonstrate the Lon protease released from dead cells catabolises co-released proteins to small peptide nutrients. Further, we suggest how such post-mortem protein lysate processing could have evolved.

[1]Department of Biosciences, Durham University, South Road, Durham DH1 3LE, UK. [2]EdinOmics, RRID:SCR_021838, School of Biological Sciences, The University of Edinburgh, Max Born Crescent, Edinburgh EH9 3BF, UK. [3]School of Physics and Astronomy, The University of Edinburgh, Peter Guthrie Tait Road, Edinburgh EH9 3FD, UK. [4]Department of Biology, University of Oxford, 11A Mansfield Road, Oxford OX1 3SZ, UK. [5]Biophysical Sciences Institute, Durham University, South Road, Durham DH1 3LE, UK. ✉e-mail: m.j.cann@durham.ac.uk

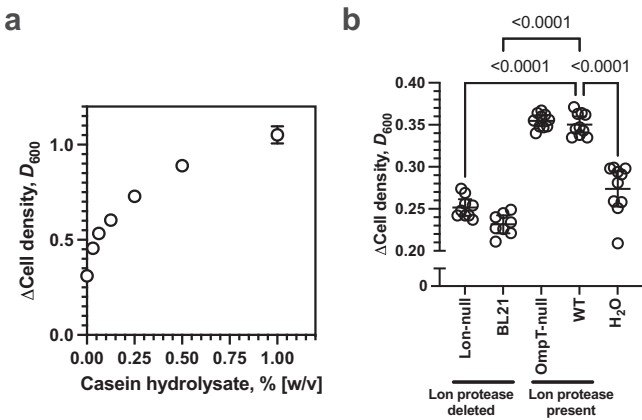

**Fig. 1 | Lon protease derived from dead bacteria is required for nutrient recycling. a** Plot of the change in cell density for *E. coli* BW25113 grown at 37 °C in M9/1% (v/v) glycerol media for 20 hrs against varying addition of casein hydrolysate ($n = 6$, mean ± 95% C.I.). **b** Plot of the change in cell density of *E. coli* BW25113 grown at 37 °C in M9/1% (v/v) glycerol media for 20 hrs in the presence or absence of lysate derived from the indicated *E. coli* strains (Lon-null, $n = 9$; BL21, $n = 8$; OmpT-null, WT, $H_2O$, $n = 10$; mean ± 95% C.I., $p$ values - one-way ANOVA with post hoc Tukey's multiple comparisons test, F (DFn = 4, DFd = 42) = 100.3). WT vs Lon-null, WT vs BL21, and WT vs $H_2O$ $p < 0.0001$; WT vs OmpT-null $p = 0.9788$; Lon-null vs $H_2O$ $p = 0.056$. Source data are provided as a Source Data file.

## Results

### Lon protease increases post-mortem nutrient recycling

We used *E. coli*-derived dead cell material (lysates), where death was induced by lysis via sonication, to investigate the underlying mechanism by which released proteins are catabolised and utilised by other live bacteria. Lysis via sonication was used as it was immediate and did not permit physiological changes in the bacteria that might be induced by other, slower-acting methods of inducing cell death. Such physiological responses, while normal for a bacterial response to stress, might lead to ambiguity in the interpretation of findings and so are avoided in this study. The sterile lysates derived from killed bacteria were examined for their growth enhancement of live cells by monitoring culture $D_{600}$ in real-time, and the data is presented as the change in cell density 20 h after the point of addition. We confirmed, through a statistical analysis of observed regression slopes, that different bacterial lysates had no impact on calibration curves of $D_{600}$ against colony-forming units (CFU) (Fig. S1) for the different live bacterial strains used in this study. Therefore, while this study is measuring bacterial biomass ($D_{600}$) and not reporting CFU mL$^{-1}$, we infer that the biomass increase is likely to represent a real increase in cell number.

Remarkably, we observed that the ability of live *E. coli* to utilise a lysate derived from dead bacteria depended on the genotype of those dead bacteria. A wild type *E. coli* BW25113 (WT) lysate enhanced live *E. coli* growth compared to the $H_2O$ control (Fig. 1b, Supplementary Fig. S2). *E. coli* BL21 is a bacterial strain commonly used for recombinant protein expression as it is ablated for genes encoding the Lon and OmpT proteases that might otherwise increase degrade the proteins being produced[6]. Therefore, we examined whether it might also have a role in post-mortem protein turnover. An *E. coli* BL21 lysate did not enhance live WT cell growth. Therefore, we hypothesised that either Lon or OmpT, or both proteases, are responsible for enhancing growth on bacterial lysates. An *E. coli* BW25113 Δ*ompT* lysate (OmpT-null) enhanced *E. coli* growth compared to the $H_2O$ control, but an *E. coli* BW25113 Δ*lon* lysate (Lon-null) did not. Lon protease has a beneficial role in stress responses in live bacteria[7]. It is an energy-dependent AAA + (ATPases associated with cellular activities) protease that performs intracellular ATP-regulated protein degradation to recognise, unfold,

translocate, and degrade substrates[8]. Lon clears more than 50% of all misfolded proteins[9] and our observation suggests an additional extracellular role in nutrient recycling.

It is possible that Lon-null lysates accumulate a toxic molecule that inhibits growth relative to wild type lysates. We exclude the possibility that the Lon protease in a bacterial lysate degrades an otherwise toxic molecule that inhibits growth, as bacterial growth is indistinguishable between live cultures grown with $H_2O$ and a Lon-null lysate (see $H_2O$ vs Lon-null comparisons in Figs. 1b, 2a, 3a, 4a, S3G, H, S5B, C). We note that there is a significant difference between $H_2O$ and Lon-null in Fig. S2A. However, this result is likely a false positive given a confidence level of 0.05 and the large number of independent observations where the two samples are statistically indistinguishable. Therefore, we infer that bacterial growth using the macromolecules released from dead bacteria depends on a phenotype (Lon protease activity) that manifests after death. Further, the observations are not likely explained by a non-specific reduction in lysate protease activity as only ablating *lon*, and not *ompT* impacts growth i.e., there is no apparent redundancy between these proteases.

We hypothesised that Lon protease, derived from dead bacteria, must catabolise proteins co-released after death to enable their use by other live bacteria. We first performed experiments to confirm that the observed post-mortem phenotype depended on the *lon* gene. Complementation restores a normal phenotype to mutants with an observable defect. We complemented the Lon-null cells to be killed and used for lysate production with a plasmid-encoded wild type Lon protease to confirm that the phenotype was due to the Lon-null genotype and not an alternative second site mutation (Fig. 2a, Supplementary Fig. S3G, H). Western blotting demonstrated that the Lon protease levels produced from the complementing plasmid were equivalent to endogenous levels in wild type *E. coli* (Supplementary Fig. S3A–F).

### Lon proteolytic activity, but not ATPase activity, is required for post-mortem nutrient recycling

Each Lon monomer contains three sub-regions: the *N*-terminal domain, the AAA+ ATPase module, and the *C*-terminal serine protease domain. Lon uses ATP hydrolysis coupled to conformational changes to unfold substrates (if necessary) and translocate the polypeptide chain to the degradation chamber[10,11]. Therefore, we investigated the dependence of the observed post-mortem phenotype on its proteolytic and ATPase activities. We complemented the Δ*lon* cells destined for lysate production with a plasmid-encoded Lon protease defective in protease activity (S679A; protease-null), ATPase activity (K362Q; ATPase-null), or both activities (S679A/K362Q; protease/ATPase-null)[12]. Lysate complemented with plasmid-derived protease-null Lon and protease/ATPase-null Lon did not rescue the Lon-null phenotype (Fig. 2a, Supplementary Fig. S3G, H). However, lysate complemented with plasmid-derived ATPase-null Lon did rescue the Lon-null phenotype, demonstrating that the role of Lon protease in post-mortem growth enhancement is ATP-independent. This observation is significant as it demonstrates an ATP-independent role for Lon, in contrast to its known ATP-dependent roles in the live cell. Further, the observation is consistent with previous findings that Lon protease has 10-30% of its full activity without ATP[13,14].

The complementation analysis proves a role in the growth enhancement phenotype for the *lon* gene, encoded in the bacterium's genome before killing. In addition, the complementation analysis demonstrates that a mutation known to impact on Lon protease activity will not complement a Lon-null phenotype. Therefore, we hypothesised that the observed protease activity in bacterial lysates would correlate with the presence of Lon protein (whether wild type or otherwise). This, we investigated whether the killed Lon-null cells are defective in a measurable protease activity compared to killed WT cells. Casein fluorescein isothiocyanate (FITC-casein) is a Lon protease substrate, and Lon protease-dependent casein degradation can be monitored indirectly by observing FITC fluorescence[15]. Therefore, we

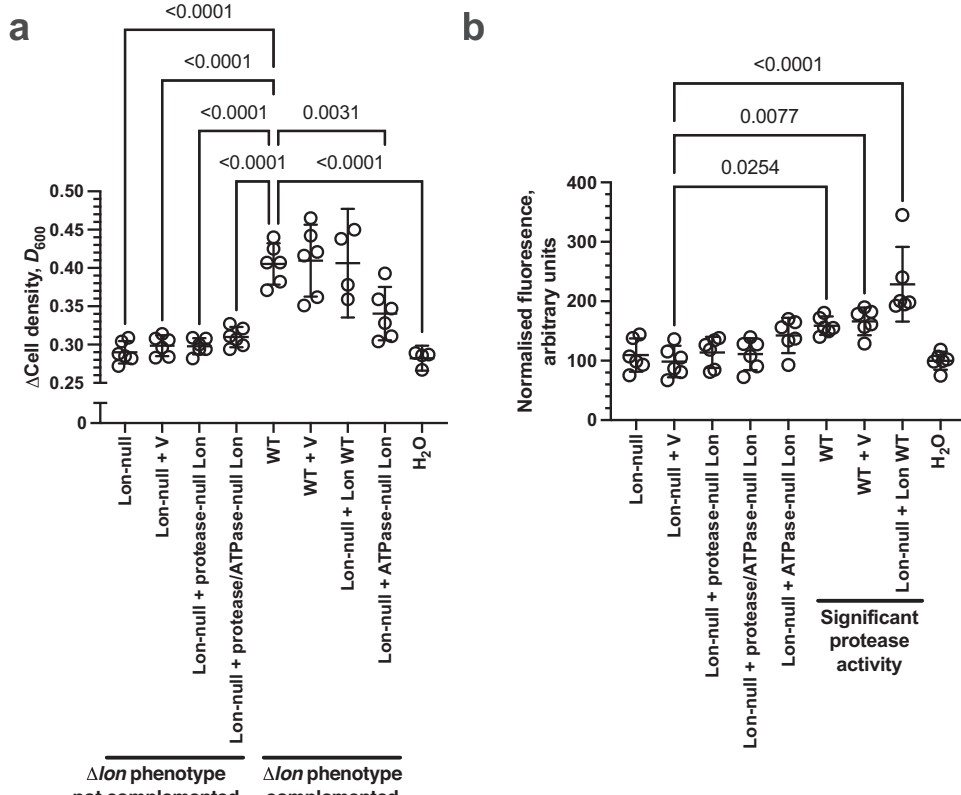

**Fig. 2 | Nutrient recycling is dependent on Lon protease activity. a** Plot of the change in cell density of *E. coli* BW25113 grown at 37 °C in M9/1% (v/v) glycerol media for 20 hrs in the presence or absence of lysate derived from WT or Lon-null cells and the indicated complementing empty plasmid (V; pBAD33) or Lon WT or mutant protein encoding plasmids (WT, WT + V, Lon-null + ATPase-null Lon, Lon-null + protease-null Lon, Lon-null + protease/ATPase-null Lon, Lon-null, Lon-null + V, $n = 6$; H$_2$O, Lon-null + Lon-WT, $n = 4$; mean ± 95% C.I., $p$ values - one-way ANOVA with post hoc Tukey's multiple comparisons test, F (DFn = 8, DFd = 41) = 23.00). Comparisons for which the Lon-null phenotype is not complemented (WT vs Lon-null, WT vs Lon-null + V, WT vs Lon-null + protease-null Lon, WT vs Lon-null +

protease/ATPase-null Lon, WT vs H$_2$O $p < 0.0001$, Lon-null vs H$_2$O $p > 0.999$). Comparisons for which the Δ*lon* phenotype is complemented (WT vs Lon-null + ATPase-null Lon $p = 0.0031$, WT vs WT + V, WT vs Lon-null + Lon WT, $p > 0.999$). **b** Plot of normalised casein-FITC fluorescence in lysate derived from *E. coli* BW25113 WT (WT) or BW25113Δ*lon* (Δ*lon*) and the indicated complementing empty plasmid (V; pBAD33) or Lon WT or mutant protein encoding plasmids ($n = 6$; mean ± 95% C.I., $p$ values - one-way ANOVA with post hoc Tukey's multiple comparisons test, F (DFn = 8, DFd = 45) = 12.39). Comparisons for significant protease activity (Lon-null + V vs WT $p = 0.0254$; Lon-null + V vs WT + V $p = 0.0077$; Lon-null + V vs Lon-null + Lon WT $p < 0.0001$). Source data are provided as a Source Data file.

measured FITC fluorescence of lysates derived from WT, Lon-null, and plasmid complemented Lon-null cells provided with FITC-casein to confirm a protease activity in these samples (Fig. 2b). Only WT lysate or lysate derived from ATPase-null Lon cells complemented with a plasmid producing WT Lon protease gave fluorescence significantly above background, demonstrating that bacterial lysates that contain presumed functional Lon protein also possess a measurable protease activity. No measurable protease activity was observed in lysates derived from Δ*lon* cells complemented with plasmids producing protease-defective protease-null Lon, ATPase-null Lon, or protease/ATPase-null Lon. Although the observation of no significant measurable protease activity with lysates derived from Δ*lon* cells complemented with plasmid producing Lon-K362Q would seem to conflict with the finding that the ATPase-null Lon lysate can rescue the Lon-null lysate phenotype, it is most likely that activity in this lysate is within the error of the experiment as a power calculation shows that the sample size is sufficient to observe a difference. Therefore, the ability of Lon protease to provide a post-mortem growth enhancement is correlated with the measurable protease activity in the bacterial lysate.

### The role for Lon protease in post-mortem nutrient recycling is not species specific

It is formally possible that Lon protease is both released from the dead cell (and, thus, present in the bacterial lysate) and secreted by the live

cells (whose growth is being enhanced) to produce the observed phenotype. To exclude any role for Lon protease produced in the live cells used in the experiments, we performed additional experiments using Lon-null cells as the live culture (Fig. 3a, Fig. S4A, B). Observations of the role of the genotype of the dead cells from which the growth-enhancing lysate was derived were identical when Lon-null cells were used as the live culture (Fig. 3a), or WT cells were used as the live culture (Fig. 2a).

We investigated whether the role of bacterial lysate-derived Lon protease in enhancing bacterial growth was restricted to only *E. coli* as the live culture. Observations of the role of the genotype of the cells from which the growth-enhancing lysate was derived were identical when *Bacillus subtilis* cells were used as the live culture (Fig. 3b, Fig. S4C, D) or *E. coli* cells were used as the live culture (Fig. 2a). Therefore, the role of Lon protease derived from dead bacteria in enhancing the growth of live bacteria is independent of Lon protease derived in the live bacteria and is not restricted to *E. coli* as the live bacteria.

### Lon protease can function after death to mediate nutrient recycling

Thus far, we have demonstrated a role for Lon protease derived from dead bacteria in enhancing live cell growth. However, we cannot exclude the possibility that Lon protease functions in the live cell. For example, it

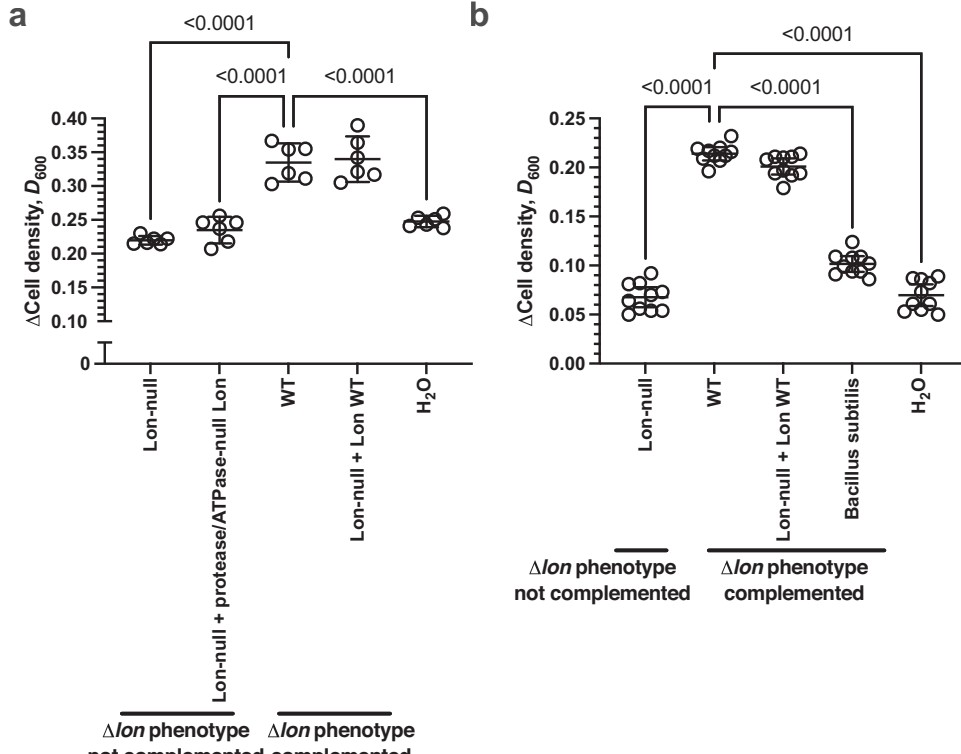

**Fig. 3 | Nutrient recycling is independent on Lon protease activity in the live cell. a** Plot of the change in cell density for Lon-null cells grown at 37 °C in M9/1% (v/v) glycerol media for 20 hrs in the presence or absence of WT or Lon-null lysates and the indicated Lon WT or mutant protein encoding plasmids ($n = 6$; mean ± 95% C.I., $p$ values - one-way ANOVA with post hoc Tukey's multiple comparisons test, F (DFn = 4, DFd = 25) = 44.50). Comparisons for which the Δ*lon* phenotype is complemented (WT vs Lon-null, WT vs Lon-null + protease/ATPase-null Lon, WT vs H₂O $p < 0.0001$; H₂O vs WT, H₂O vs Lon-null + Lon WT $p < 0.0001$). Comparison for which the Δ*lon* phenotype is not complemented (Lon-null vs H₂O $p = 0.1869$). **b** Plot of the

change in cell density for *Bacillus subtilis* grown at 30 °C in M9/1% (v/v) glucose media for 20 hrs in the presence or absence of WT, Lon-null, or *B. subtilis* lysates and the indicated Lon WT protein-encoding plasmid ($n = 10$; mean ± 95% C.I., $p$ values - one-way ANOVA with post hoc Tukey's multiple comparisons test, F (DF = 4, DFd = 45) = 324.1). Comparisons for which the Δ*lon* phenotype is complemented (WT vs Lon-null, WT vs *Bacillus subtilis*, WT vs H₂O $p < 0.0001$; H₂O vs Lon-null + Lon WT $p < 0.0001$). Comparison for which the Lon-null phenotype is not complemented (Lon-null vs H₂O $p = 0.9957$). Source data are provided as a Source Data file.

is possible that Lon protease produces nutrients in the live cell that have a post-mortem function after release from dead cells. A purified recombinant Lon protease can be used to add Lon activity to the lysates of killed Lon-null cells. This approach enables us to separate the post-mortem Lon function from the function of Lon produced in the live cell. Therefore, we complemented the Lon-null lysate phenotype with recombinant Lon protease added to the lysate after cell killing. The amount of recombinant Lon protease added to lysates was equivalent to the levels of endogenous Lon protease observed in lysates of WT cells (Fig. S3A–F). Measurable protease activity was detected in lysates containing appropriate recombinant WT Lon protease concentrations (Fig. S5A). The protease activity from lysates supplemented with recombinant protease-null Lon or protease/ATPase-null Lon was indistinguishable from the addition of buffer alone. Adding either wild type Lon protease or ATPase-null Lon protein complemented the Lon-null lysate phenotype (Fig. 4a, Fig. S5B, C). The protease activity from lysates supplemented with recombinant ATPase-null Lon was between that of wild type Lon protease and buffer alone, reflecting the lower specific activity of this enzyme. No complementation was observed with protease-null Lon protein or the protease/APTase-null Lon protein. Therefore, Lon protease's function in nutrient recycling is post-mortem and most likely not dependent on its presence in the cell before death. This conclusion is further backed by our result above that demonstrates that the post-mortem function is ATP-independent, in contrast to the ATP-dependent function of Lon in the live cell (Fig. 2a). However, while recombinant Lon protease complements the Lon-null phenotype fully,

we cannot formally exclude the possibility that Lon protease acting in the live cell before killing has some contribution to the total nutrient pool that contributes to growth.

Recombinant Lon proteases can be used as a further tool to investigate compatibility between the growing bacterial species and the source of the Lon protease. Therefore, we investigated whether the Lon protease post-mortem function was specific to the *E. coli* enzyme or a more general property of the enzyme. Further, Lon orthologues are divided into two subgroups[16]; A type Lon proteases, including that from *E. coli*, which have a multi-lobed *N*-terminal domain together in conjunction with the ATPase and protease domains, and B type Lon proteases, found in the Archaea, which lack the *N*-terminal domain. Thus, we investigated the ability of the B type Lon of *Thermococcus onnurineus* NA1[17] (*Ton*Lon) to complement the Lon-null lysate phenotype (Fig. 4b, Fig. S5D). The effect of recombinant *Ton*Lon was indistinguishable from that of recombinant *E. coli* Lon protease in complementing the Lon-null lysate phenotype, confirming that the role of Lon protease is not species, or Lon type, specific. Therefore, while a role in nutrient recycling is not a generic property of all proteases (e.g., see OmpT in Fig. 1b), it is likely a generic property of all Lon proteases.

### Lon protease is required for small peptide production in post-mortem nutrient recycling

A role for Lon protease in enabling post-mortem nutrient utilisation and the observation that this phenotype correlates with the protease activity present in the lysates of killed bacteria indicates that Lon

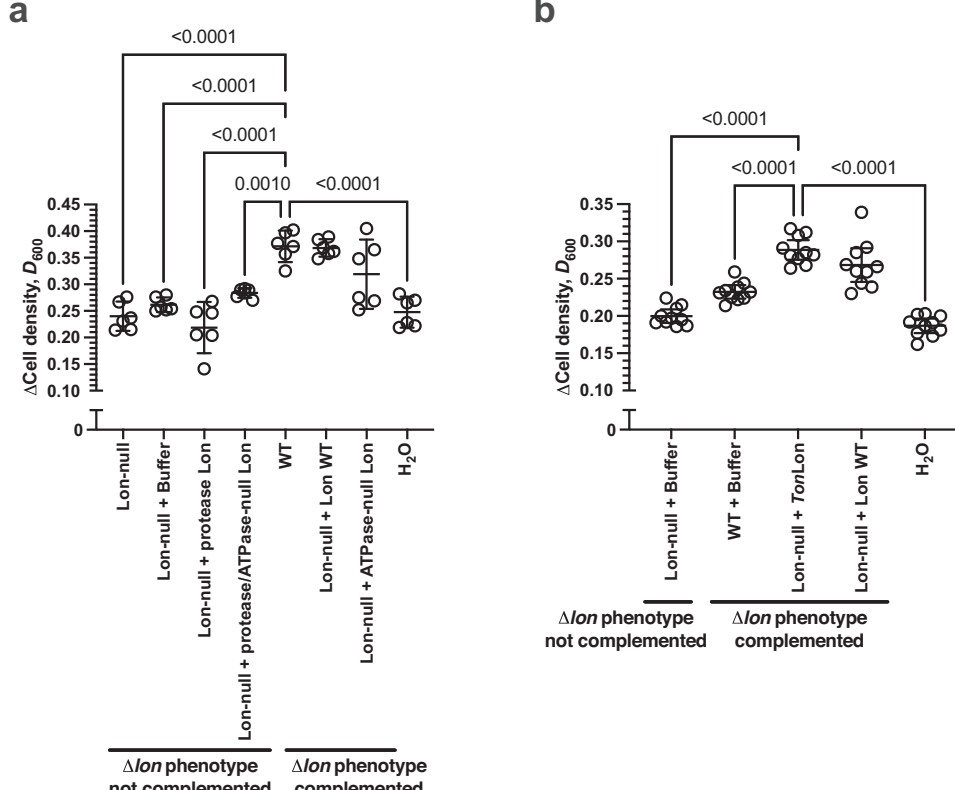

**Fig. 4 | Nutrient recycling is dependent on post-mortem Lon protease activity.** Plots of the change in cell density for *E. coli* BW25113 grown at 37 °C in M9/1% (v/v) glycerol media for 20 hrs in the presence or absence of lysate derived from *E. coli* BW25113 WT (WT) or BW25113Δ*lon* (Δ*lon*) and the indicated recombinant Lon proteins. **a** Wild type and mutant *E. coli* Lon protease ($n = 6$; mean ± 95% C.I., $p$ values - one-way ANOVA with post hoc Tukey's multiple comparisons test, F (DFn = 7, DFd = 40) = 18.81). WT vs Lon-null, WT vs Lon-null + buffer, WT vs protease-null Lon, WT vs H$_2$O $p < 0.0001$; WT vs protease/ATPase-null Lon $p = 0.0010$; WT vs ATPase-null Lon $p = 0.1349$; Lon-null vs H$_2$O $p > 0.9999$. **b** Wild type *E. coli* Lon protease and *Thermococcus onnurineus* NA1 Lon protease (*Ton*Lon) ($n = 10$; mean ± 95% C.I., $p$ values - one-way ANOVA with post hoc Tukey's multiple comparisons test, F (DFn = 4, DFd = 45) = 51.51). H$_2$O vs WT + buffer, H$_2$O vs Lon-null + *Ton*Lon, H$_2$O vs Lon-null + Lon WT $p < 0.0001$; H$_2$O vs Lon-null + buffer $p = 0.5634$. Source data are provided as a Source Data file.

specifically catabolises high molecular weight intact proteins to smaller molecular weight peptides for transport into the live bacteria. Bacteria possess numerous peptide transporters for importing peptides as nutrients. There are two major categories of peptide transporters. First, the proton motive force-driven transporters (POT or PTR family), which cluster in pairs of YdgR/YhiP and YjdL/YbgH[18] and utilise proton import for transport. Second, the ATP binding cassette-containing transporters (ABC transporters) that include the dipeptide permeases (DdpABCDF and DppABCDF) and the oligopeptide permease (OppABCDF)[19]. The most common peptide substrates for uptake are di- and tripeptides.

Given the role of Lon protease in post-mortem nutrient recycling and its link to measured protease activity in lysates derived from dead bacteria, we hypothesised that Lon protease produces peptides for uptake by peptide transporters. First, we assessed peptide levels inside bacterial cells by measuring their concentration immediately after release from the cell (Fig. 5a). We observed, no significant differences between all the assessed strains (other than a single significant difference between Lon-null cells complemented with plasmid producing protease/ATPase-null Lon or WT Lon). Importantly, there was no difference between WT and Lon-null cells indicating that WT cells do not contain higher peptide levels per se.

The method for peptide quantitation can detect peptides in the 6–10 amino acid range. Therefore, we assessed peptide catabolism post-mortem by observing peptide loss in this range (Fig. 5b). We measured the ratio of 6–10 amino acid peptides after and before incubation post-mortem for 20 h at 37 °C where ratio <1 indicates peptide catabolism.

Only strains producing WT Lon or ATPase-null Lon showed significant peptide catabolism post-mortem. This finding is consistent with the observation that extracellular Lon will degrade casein-FITC (Fig. 2b) with the additional caveat that this experiment was sufficiently sensitive to observe a catalytic activity in ATPase-null Lon, consistent with its ability to complement the Lon-null phenotype (Figs. 2a, 4a).

WT and Lon-null cells contain equivalent peptide levels in the 6-10 amino acid range but only Lon-containing cells catabolise those peptides post-mortem. Therefore, we used used liquid chromatography coupled to ion mobility quadrupole time-of-flight mass spectrometry to address di- and tri-peptide levels in WT and Lon-null lysates suitable for uptake into bacteria for use as nutrients. First, we assessed di- and tri-peptide levels inside bacterial cells by measuring their concentration immediately after release from the cell (Fig. 5c). We observed no difference in the distribution of 923 amino acids, di- and tri-peptides between WT and Lon-null lysates. We then measured the change in amino acids, di- and tri-peptides after lysate incubation post-mortem for 20 h at 37 °C (Fig. 5d). We observed a significant increase in the distribution of 923 amino acids, di- and tri-peptides in WT compared to Lon-null lysates.

We examined the top 30 di- and tripeptides contributing to the differences between biologically independent WT and Lon-null lysates presented as variable importance in projection scores (Fig. 5e, Fig. S6). 23 of the 30 differential features are higher (group average) in the WT compared to the Lon-null groups ($p = 0.0026$, one-tailed binomial hypothesis test), consistent with the increase in distribution of amino acids, di- and tri-peptides in the WT lysate. We

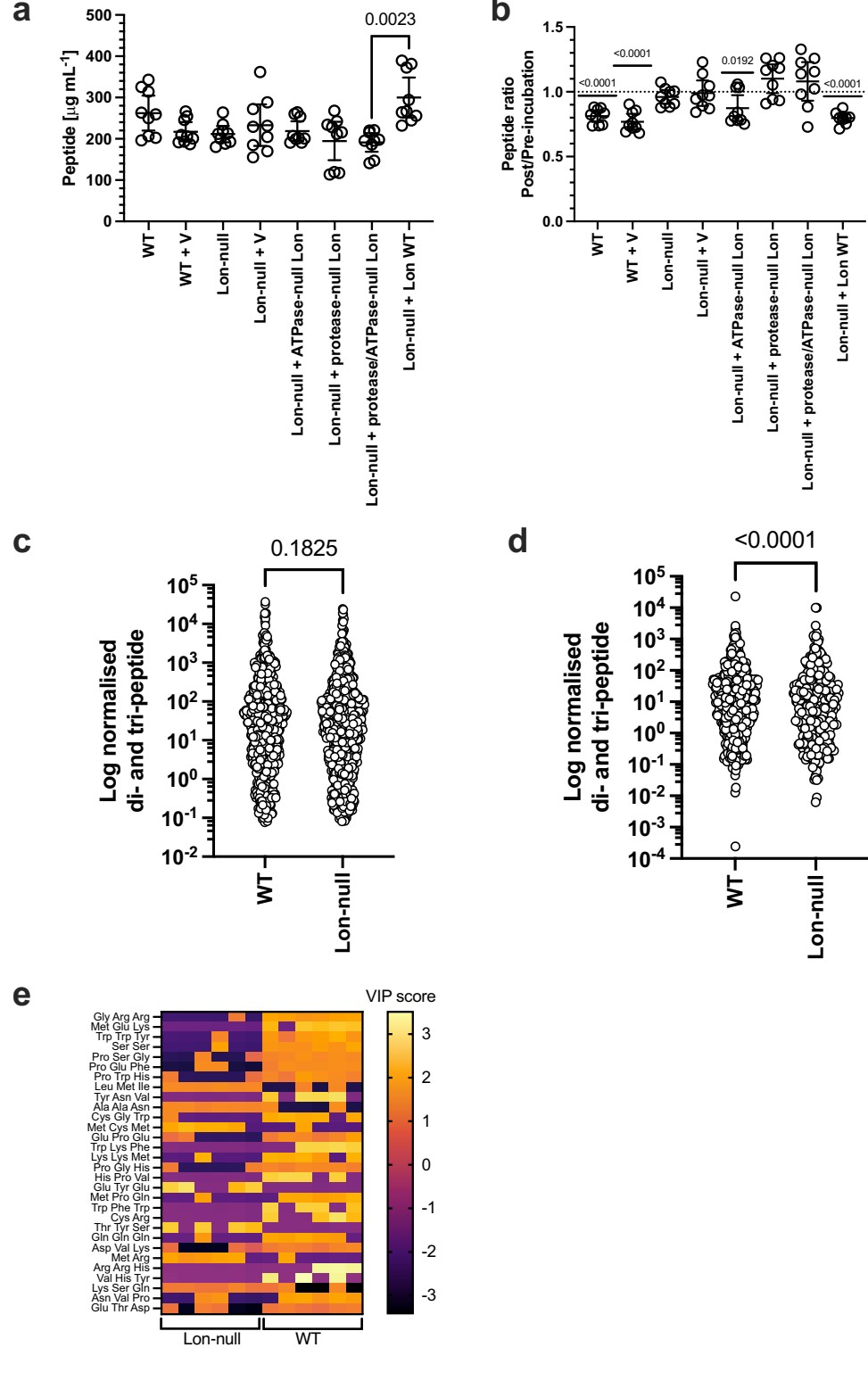

cannot eliminate the possibility that there is a Lon contribution to a subset of peptides found in live cell that are released post-mortem and contribute to nutrient recycling. However, we have demonstrated that Lon protease does have a significant role in generating peptides of the appropriate size for uptake by peptide transporters as nutrients, thus providing further evidence for its role in nutrient recycling post-mortem.

## Lon protease appears to be a post-mortem social adaptation

We next asked why natural selection would have favoured a post-mortem phenotype. We have already shown that Lon protease provides a post-mortem benefit to other cells (Fig. 1b). Lon protease provides this benefit by breaking down proteins into di- and tri- peptides, which the surrounding cells can utilise for enhanced growth. This suggests that Lon protease is a cooperative trait that provides a

**Fig. 5 | Lon protease is required for small peptide production in *E. coli* lysate.** **a** Plot of the peptide content of the indicated bacterial strains ($n = 9$, mean ± 95% C.I., p-values−Kruskal−Wallis multiple comparisons test with post hoc Dunns multiple comparisons test, Kruskal-Wallis statistic = 21.89). **b** Plot of the ratio of the peptide content of the indicated bacterial strains before and after incubation at 37 °C ($n = 9$, mean ± 95% C.I., p values - one-sample two-tailed Wilcoxon Test, hypothetical mean = 1). **c** Log plot of the distribution of 923 amino acids, di- and tri-peptides quantities (normalised to total sample protein) for WT and Lon-null cells directly after cell death (two-tailed Kolmogorov-Smirnov test, Kolmogorov-Smirnov D = 0.05092). **d** Log plot of the difference in distribution of 923 amino acids, di- and tri-peptides quantities (normalised to total sample protein) for WT and Lon-null cells after incubation at 37 °C (two-tailed Kolmogorov-Smirnov test, Kolmogorov-Smirnov D = 0.1463). **e** The plot is a heat map for the top thirty features contributing to the differences between WT and Lon-null post-mortem bacterial lysates presented as variable importance in projection scores ($n = 6$). Source data are provided as a Source Data file.

benefit or public good to other cells. Consequently, If Lon protease production is costly then it could potentially be exploited by individuals who do not produce it, termed cheats[20].

**Lon protease production is costly and exploitable.** We examined whether Lon production is costly by growing over 20 h (early stationary phase) at 37 °C in minimal media with a single carbon source. We found that Lon-null mutant cells grew to a higher density than both WT cells and Lon-null cells complemented with a plasmid carrying the Lon protease, which do produce Lon protease (Fig. 6a). Therefore, a fitness cost is associated with Lon protease production at a growth phase before significant cell death.

We then examined if Lon protease production could be exploited by cheats[20]. We tested this possibility by growing a mixed culture of non-producers (cheats; Lon-null) and WT cells that produced Lon protease for a period (5 days) over which cell death occurred thus making the Lon protease producing cells exploitable. Over 5 days of culture at 37 °C the non-producing cheats increased in frequency from 3% to 12%, supporting the hypothesis that they can exploit the Lon protease production of the WT (Fig. 6b). Importantly, this experiment was performed with natural cell death at stationary phase, not induced cell death by sonication.

Social evolution theory also predicts that the fitness of cheats will depend upon their frequency in the population, with the relative fitness of cheats being higher when they are rarer[21]. When cooperators are more common, there is greater population growth and more opportunity for cheats to exploit the cooperators. In support of this prediction, we found that the relative fitness of Lon protease non-producers (Lon-null), was lower when they were more common, after 5 days of culture at 37 °C (Fig. 6c). Overall, these results support the hypothesis that the production of Lon protease is a cooperative public good that can be exploited by cheats that do not produce it.

**Public goods and kin selection.** The production of public goods can be favoured when they are directed towards relatives who share the gene for producing the public good (kin selection)[22]. One way of conceptualising this is that a high relatedness prevents cheats from exploiting cooperators, because they will be interacting with other cheats and not cooperators. Other examples of public goods produced by bacteria include iron scavenging siderophore molecules, elastase and crystal toxins[20,23,24]. Support for a role of kin selection in bacteria has been provided by laboratory experiments, across species studies and population genetic analyses[20,23,25−27]. Kin selection is likely to be of widespread importance in bacteria where clonal growth leads to neighbouring and interacting cells being close relatives (clone mates).

**Lon protease can provide a private benefit.** Under certain conditions, the production of Lon protease could potentially also provide a direct (private) benefit to living cells through protein quality control, as it degrades the majority of unfolded proteins that arise under stress (e.g., elevated temperature)[28]. We tested this possibility by examining the fitness consequences of producing Lon protease under heat shock conditions (42 °C) that induce stress, but with negligible cell death (so no public benefit)[29]. We reasoned that heat shock conditions would enhance any observations of a private benefit for Lon protease.

Under stressful heat shock conditions, we found that Lon-null cells grew to a lower density than WT cells or Lon-null cells complemented with a plasmid carrying the Lon protease - when grown for 20 h at 42 °C in minimal media with a single carbon source cell (Fig. 7a). This suggests that Lon protease production can provide a benefit to the live cell other than the post-mortem public benefit. We then investigated how these benefits could sum by growing monocultures under stress at the late stationary phase when significant cell death occurs. Cultures were initiated at day 0 with an equivalent number of cells, and CFU mL$^{-1}$ was determined after 5 days of culture at 42 °C (Fig. 7b). Lon-null cells were absent after 5 days, indicating that the fitness cost of not producing Lon protease can lead to complete population collapse when significant cell death occurs.

As a further test of whether the benefit of Lon in living cells was private, we tested for frequency dependence. We grew Lon-null cells in mixed cultures for 20 h, corresponding to late-log phase and so prior to significant cell death and any Lon release as a public good. Experiments were performed at 37 °C (no stress and no Lon private function required) and 42 °C (under stress and Lon private function required) (Fig. 7c, d). In both cases, we observed no correlation between relative fitness of the Lon-null non-producing mutants and their percentage in the starter culture at either temperature. This further supported the hypothesis that the benefit of Lon in living cells is not a cooperative trait that can be exploited by cheats.

**Private benefit does not outweigh cost.** We then asked whether the private benefit of producing Lon protease can completely outweigh the cost under our examined experimental conditions, such that the trait provides a net benefit at the private level. If this was the case, then the cooperative benefit of Lon protease could potentially be explained as a byproduct of a private trait, rather than a social adaptation. We initiated Lon protease producing WT populations with a small proportion (~3%) of non-producing mutant (Lon-null) cells and followed their growth after 5 days of culture at 42 °C (Fig. 7e). The non-producing Lon-null cheats showed a small but significant increase in population frequency over this time, suggesting that they can still exploit producers. Consequently, the private benefit under stressful conditions does not outweigh the cost – the production of Lon protease is still net costly.

While the method of inducing cell death (sonication) in defining the post-mortem Lon function differs from the natural cell death occurring in these social adaptation studies, the findings are congruent. All the same, we cannot eliminate the possibility of a hitherto undefined physiological response in addition to Lon in natural cell death. Nonetheless, taken together, our results suggest that Lon protease is a cooperative public good - a social adaptation that is likely to be partially explained by kin selection. Although it can also provide a private benefit under stressful conditions, we found that this private benefit does not outweigh the cost under our examined experimental conditions. This quantitative conclusion is for the conditions we examined, and the relative importance of private and public benefits could vary with environmental conditions, such as the relative occurrence of stressful conditions, and the potential gains from post-mortem nutrient utilisation. In the extreme, different scenarios could lead to the private benefit being either negligible or outweighing the cost.

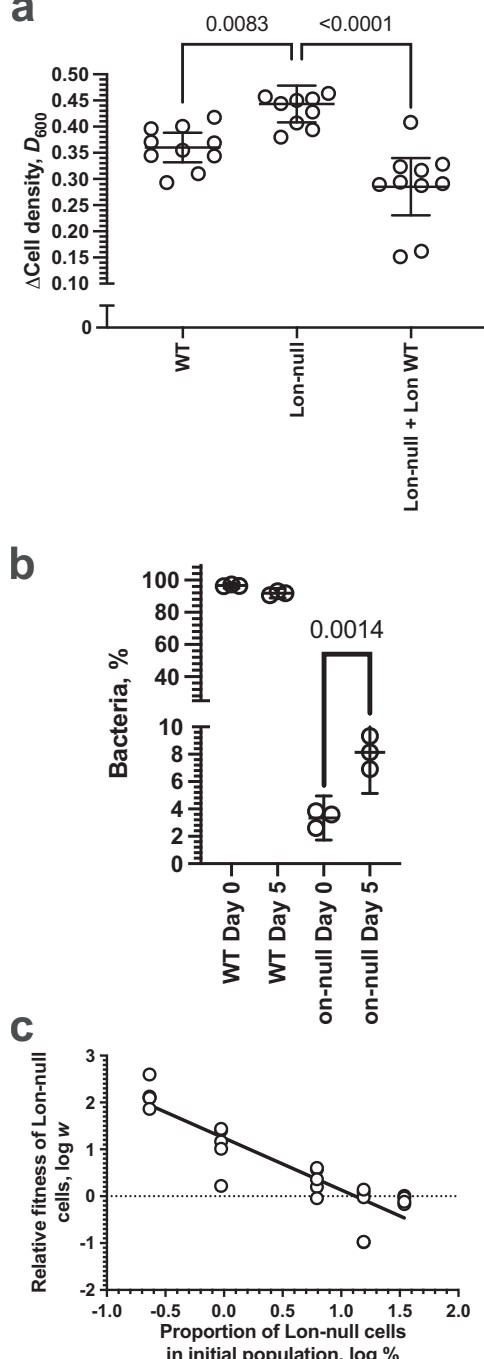

**Fig. 6 | Lon protease production is costly and exploitable. a** Plot of the change in cell density for the indicated WT or Lon-null strains with or without plasmid encoding WT Lon protein grown at 37 °C in M9/1% (v/v) glycerol media for 20 h ($n = 10$; mean ± 95% C.I., $p$-values—one-way ANOVA with post hoc Tukey's multiple comparisons test, F (DFn = 2, DFd = 27) = 19.18). WT vs Lon-null $p = 0.0083$; WT vs Lon-null + Lon WT $p = 0.0179$; Lon-null vs Lon-null + Lon WT $p < 0.0001$. **b** Plot of the percentage of the indicated WT or Lon-null strains before and after growth in a mixed culture at 37 °C in M9/1% (v/v) glycerol media for five days ($n = 3$; mean ± 95% C.I., $p$ values - one-way ANOVA with post hoc Tukey's multiple comparisons test, F (DFn = 3, DFd = 8) = 8792). WT Day 0 vs Day 5, Lon-null Day 0 vs Day 5 $p = 0.0014$. **c** Plot of the relative fitness of Lon-null bacteria at 37 °C grown for 5 days, compared to WT cells, against the proportion of cells in the initial population ($n = 3$, $R^2 = 0.8095$, Spearman $r = -0.90$ $p = 0.0417$). Source data are provided as a Source Data file.

The private benefit under stressful conditions makes it easier for kin selection to explain Lon protease relative to a purely altruistic cooperative trait. The ability to cheat was much reduced under heat shock conditions and this ability to exploit producers did not show frequency dependence (after 5 days of culture at 42 °C; Fig. 7f). We observed no correlation between relative fitness of the Lon-null non-producing mutants and their percentage in the starter culture. Thus, selection for Lon protease production would require a lower relatedness than if this private benefit did not also occur, depending upon how often stressful conditions occurred.

## Discussion

We have demonstrated a post-mortem function and its role in nutrient recycling in a bacterial species, which appears to be a cooperative social adaptation. The possible importance of post-mortem functions could be much wider. For example, the cellular contents of a unicellular green algae can have differing effects on the relative growth rate of other living cells, depending on the mode of cell death to produce those contents[30] and it has been suggested that natural selection might affect the post-mortem trait of leaf litter decomposability because of the effect on soil fertility, to the benefit of the plant that dropped those leaves[31,32]. Further, gene transcription profiles in mice and zebrafish have been observed to have a significantly ordered component post-mortem, but through an unknown mechanism[33]. We hypothesise that many species may have post-mortem phenotypes that are favoured by natural selection.

## Methods

### Bacterial strains and plasmids

Unless otherwise stated, all experiments used wild type and mutant *E. coli* BW25113 strains of the Keio collection[34]. Lon protease encoding wild type and variant open reading frames (Uniprot P0A9M0) were produced by gene synthesis (Genscript). Complementation analyses were performed using Lon protease protein produced from the pBAD33 vector using background transcription without the arabinose transcriptional activator[35].

### Bacterial lysate preparation

Bacteria for lysate production were grown in M9 minimal media supplemented with 1% (v/v) glycerol and grown to $D_{600} = 0.4$ at 37 °C. Cells were centrifuged (2000 g, 20 mins), resuspended in sterile $dH_2O$, and lysed by sonication. The sonicated lysate was filtered through a 0.22 µm Steriflip™ (Millipore), flash frozen in liquid $N_2$ and stored at -80 °C. Processed lysates were checked for sterility by plating on Luria-Bertani (LB) broth agar and growth in LB broth liquid culture.

### *E. coli* bacterial growth

*E. coli* strains were grown overnight in M9 minimal media supplemented with 1% (v/v) glycerol. The bacteria were diluted to $D_{600} = 0.2$ using fresh M9 minimal media supplemented with 1% (v/v) glycerol. 50 µL of live, diluted bacteria were added to 50 µL of bacterial lysate (or $dH_2O$ for the growth-only negative control), resulting in a starting $D_{600}$ of 0.1 in all growth assays. The live bacteria were then grown for an additional 1200 mins at 37 °C with shaking in a SPECTROstar Nano™ absorbance plate reader. All bacterial lysates were incubated without live bacteria to confirm sterility. Colony-forming unit counts were performed by plating a dilution series on LB broth agar.

### *Bacillus subtilis* bacterial growth

*B. subtilis* strain 168 was grown in minimal M9 media supplemented with 1% (v/v) glucose. Growth assays were executed as per *E. coli* strains, albeit with glucose being exchanged for glycerol in all cases.

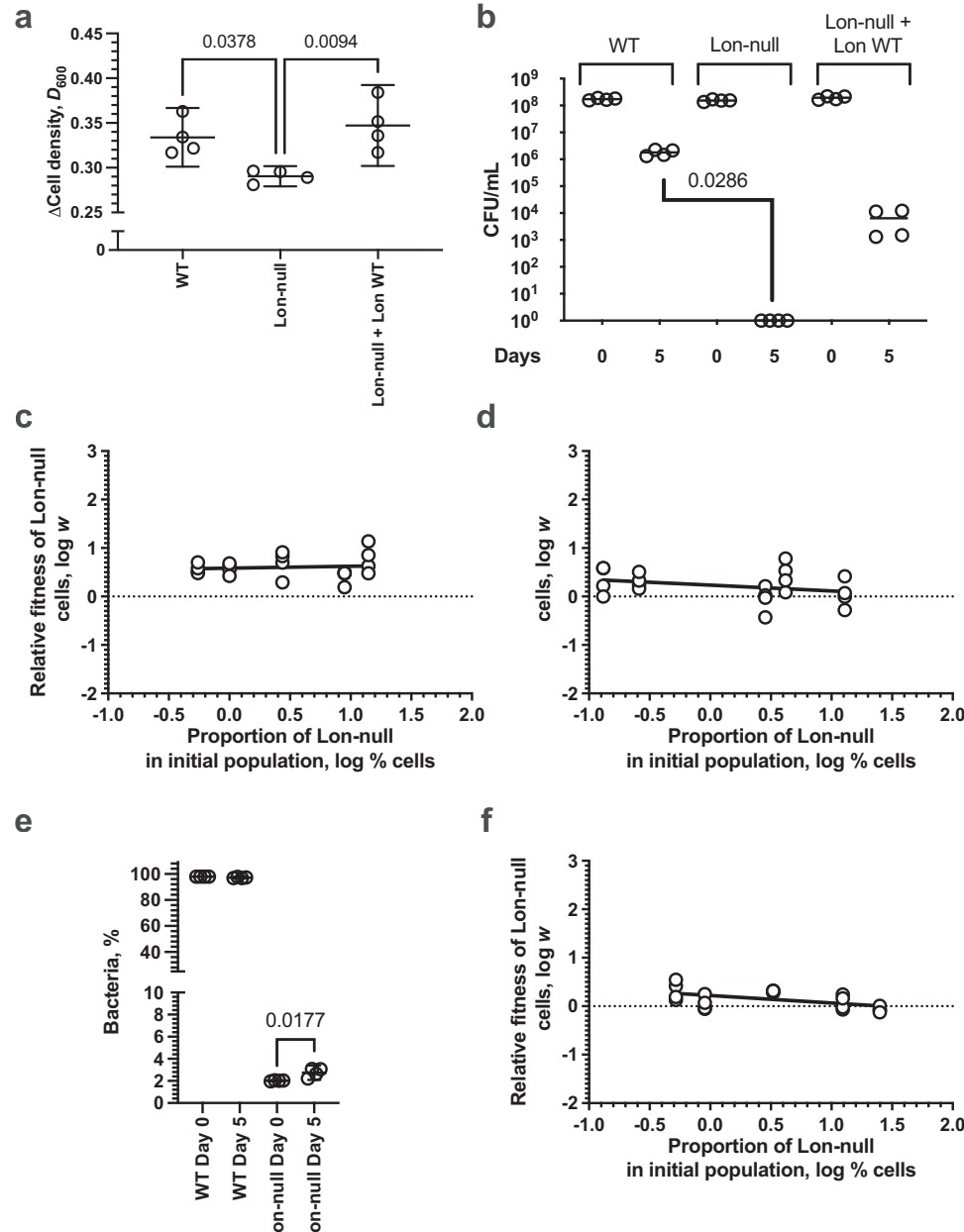

**Fig. 7 | Lon protease can provide a private benefit that does not outweigh cost.**
**a** Plot of the change in cell density for the indicated WT or Lon-null strains with or without plasmid encoding WT Lon protein grown at 42 °C in M9/1% (v/v) glycerol media for 20 hrs ($n = 4$; mean ± 95% C.I., $p$ values−one-way ANOVA with post hoc Tukey's multiple comparisons test, F (DFn = 2, DFd = 9) = 8.224). WT vs Lon-null $p = 0.0378$; WT vs Lon-null + Lon WT $p = 0.6507$; Lon-null vs Lon-null + Lon WT $p = 0.0094$. **b** Plot of the change in CFU mL$^-$ for the indicated *E. coli* BW25113 WT (WT) or Δ*lon* (Δ*lon*) strains with or without plasmid encoding WT Lon protein grown at 42 °C in M9/1% (v/v) glycerol media for 5 days ($n = 4$); mean ± 95% C.I., $p$ value−two-tailed Mann–Whitney Test. **c** Plot of the relative fitness of Lon-null bacteria at 37 °C grown for 20 hrs, compared to WT cells, against the proportion of

cells in the initial population ($n = 3$, $R^2 = 0.01027$, Spearman $r = 0.40$ $p = 0.2583$). **d** Plot of the relative fitness of Lon-null bacteria at 42 °C grown for 20 hrs, compared to WT cells, against the proportion of cells in the initial population ($n = 3$, $R^2 = 0.09516$, Spearman $r = -0.20$ $p = 0.3917$). **e** Plot of the percentage of the indicated WT or Lon-null strains before and after growth in a mixed culture at 42 °C in M9/1% (v/v) glycerol media for five days ($n = 4$; mean ± 95% C.I., $p$-values−one-way ANOVA with post hoc Tukey's multiple comparisons test, F (DFn = 3, DFd = 12) = 143155). WT Day 0 vs Day 5, Lon-null Day 0 vs Day 5 $p = 0.0177$. **f** Plot of the relative fitness of Lon-null bacteria at 42 °C grown for 5 days, compared to WT cells, against the proportion of cells in the initial population ($n = 3$, $R^2 = 0.2765$, Spearman $r = -0.70$ $p = 0.1167$). Source data are provided as a Source Data file.

## FITC-casein assay
20 µL of material was added to 40 µL of assay buffer (100 mM Tris pH 7.8 and 10 mM CaCl) and 40 µL of 0.5% (w/v) FITC-Casein. The assay was incubated for 2 h in brown microcentrifuge tubes at 37 °C with constant mixing. After incubation, 240 µL of fresh 5% (w/v) tri-chloroacetic (TCA) acid was added to stop the reaction and precipitate all protein. Tubes were mixed for 2 minutes and then stood in the dark at room temperature for 4 h. The TCA-insoluble material was pelleted

by centrifugation. 120 µL of the supernatant was diluted with 680 µL of 500 mM Tris pH 8.5. Emission spectra were taken using an excitation wavelength of 365 nm and reading emission wavelengths of 470–600 nm with an emission maximum of 525 nm.

## Western blot
1.0 mm polyacrylamide gels (10% (v/v) bis-acrylamide resolving and 5% (v/v) bis-acrylamide stacking) were poured using the Mini-PROTEAN

Tetra Electrophoresis System. Samples were mixed 1:1 (v:v) with loading buffer (50 mM Tris-HCl pH 6.8, 2% (w/v) SDS, 0.1% (w/v) bromophenol blue, 10% (v/v) glycerol, and 100 mM DTT), incubated at 95 °C for 5 minutes and run at 20 V cm$^{-1}$ in running buffer (25 mM Tris-HCl pH 6.8, 200 mM glycine, 0.1% (w/v) SDS). Proteins were transferred at 2 V cm$^{-1}$ at 4 °C overnight in transfer buffer (25 mM Tris-HCl pH 8.5, 190 mM glycine, and 15% (v/v) methanol). Membranes were washed for 5 minutes in TBS-T (25 mM Tris-HCl pH 7.5, 150 mM NaCl, 0.05% (v/v) Tween-20) and incubated in blocking buffer (5% (w/v) non-fat milk in TBS-T) for 2 h at room temperature. Membranes were washed three times in TBS-T for 10 minutes each and then probed with 1/10,000 α-Lon protease primary antibody (Sino Biological, 40219-T24) diluted in blocking buffer. Membranes were rewashed with TBS-T and then probed with 1/5,000 goat anti-rabbit secondary antibody (Abcam, ab205718) diluted in blocking buffer. Membranes were again washed with TBS-T before developing with ECL™ Western Blotting Detection Reagent at room temperature.

## Protein production

Wild type *E. coli* Lon protease[36] and the K362Q, S679A, and K362Q/S679A variants[12] were produced from cultures of *E. coli* strain BL21 carrying plasmid pBAD33-*lon*[37] grown at 37 °C in 2xYT medium supplemented with chloramphenicol (34 µg mL$^{-1}$). L-arabinose was added (0.2% (w/) at a $D_{600}$ of 1.0), and protein production was induced for 3 h before harvesting cells and freezing them at −80 °C. Cells were thawed in 25 mL of cold buffer A (100 mM potassium phosphate pH 6.5, 1 mM DTT, 1 mM EDTA, 10% (v/v) glycerol) per 2 L of initial culture, and the cell suspension was lysed by a French pressure cell press. The lysate was centrifuged (18,000 g, 30 min) and then loaded onto a HiTrap SP column preequilibrated with cold buffer A. The column was washed with 25 mL buffer A and 50 mL buffer B (200 mM potassium phosphate pH 6.5, 1 mM DTT, 1 mM EDTA, 10% (v/v) glycerol). The protein was eluted from the column using 50 mL of elution buffer (400 mM potassium phosphate pH 6.5, 1 mM DTT, 1 mM EDTA, 10% (v/v) glycerol). The protein was dialysed into 50 mM HEPES (pH 7.5), 250 mM NaCl, 1 mM DTT, 1 mM EDTA. Following concentration, the sample was loaded onto a preequilibrated 16/60 Sephacryl S300 gel filtration column (Merck). Fractions were confirmed to contain Lon protease by SDS-PAGE. The confirmed fractions were combined, dialyzed against 50 mM HEPES (pH 7.5), 250 mM NaCl, 1 mM DTT, 1 mM EDTA and 10% (v/v) glycerol, and stored in aliquots at −80 °C.

Wild type *Thermococcus onnurineus* Lon protease (*Ton*Lon) was produced by growing *E. coli* strain BL21(DE3) carrying a pET24a vector containing the TON_0529 gene[17]. Bacteria were grown in LB broth containing 50 µg ml$^{-1}$ kanamycin to a $D_{600}$ of 0.5 at 37 °C. Expression of *Ton*Lon was induced with 1 mM isopropyl-*D*-1-thiogalactopyranoside. After an 18 h expression at 20 °C, the cells were harvested and resuspended in 50 mM Tris−HCl pH 8.0 and 500 mM KCl. The cells were lysed by sonication, and the lysate was clarified by centrifugation at 18,000 g for 45 min at 4 °C.

The clarified lysate was loaded onto a nickel−nitrilotriacetic acid (HisTrap) column (Cytiva). After collecting the flow through, the column was washed with 50 mM Tris−HCl pH 8.0, 500 mM KCl and 10 mM imidazole. *Ton*Lon was eluted from the column by using a gradient of imidazole with an elution concentration of 300 mM imidazole. The elutes containing *Ton*Lon were dialysed into a SEC buffer containing 50 mM Tris−HCl pH 8.0 and 15 mM MgCl$_2$, after which the dialysed fractions were concentrated. The concentrated sample was loaded onto a Superdex 200 HR 16/60 column (Cytiva) pre-equilibrated with SEC buffer. After elution from the column, the fractions were confirmed by SDS-PAGE analysis. The *Ton*Lon fractions were dialysed into storage buffer (50 mM Tris−HCl pH 8.0, 15 mM MgCl$_2$ and 10% (v/v) glycerol) and concentrated. Aliquots of purified protein were frozen using liquid N$_2$ and stored at -80 °C.

All complementation analyses used the same pBAD33 constructs carrying the indicated *lon* variants. The empty pBAD33 plasmid is denoted as V in the text.

## Fitness assays

The relative percentage genotypes of mixed wild type and mutant cultures were calculated by plating cultures on both LB agar and LB agar containing 50 µg mL$^{-1}$ kanamycin. The CFU count on kanamycin corresponded to the Δ*lon* mutant due to the presence of the gene-inactivating kanamycin-encoding cassette. The wild type CFU count was calculated by subtracting the Δ*lon* count from the total CFU count on LB agar. We calculated the relative fitness of cheating Δ*lon* mutants (*w*) by comparing the frequency of cheats at the beginning and end of the experiment. Specifically, *w* is given by $[x_2(1 - x_1)]/[x_1(1 - x_2)]$, where $x_1$ is the initial proportion of cheats in the population and $x_2$ is their final proportion[21]. For example, $w = 2$ would correspond to the mutant's growing twice as fast as the wild-type cooperator. We used measured CFU counts for experiments that required culturing over 5 days, rather than $D_{600}$ measurements, as dead cell debris rendered $D_{600}$ measurements unreliable as a measure of live cell biomass.

## Peptide analysis

Peptides were quantified using a Pierce™ BCA peptide assay kit (Thermo Scientific). Samples were extracted in a 1:5 dilution with ice-cold chloroform/methanol/water (1:3:1) by incubating at 4 °C for 1 h at 1000 rpm. The extraction mixtures were centrifuged at 13,000 × g for 10 min, and 100 µL of the supernatant was transferred into sterile 1.5 mL microcentrifuge tubes for storage at −80 °C until analysis. The analysis of amino acids, dipeptides and tripeptides was performed using liquid chromatography (LC) coupled to ion mobility (IM) quadrupole time-of-flight (qTOF) mass spectrometry (MS)[38]. The instrumentation consisted of an Agilent 1290 Infinity II series UHPLC system hyphenated with an Agilent 6560 IM-qTOF with a Dual Agilent Jet Stream Electron Ionization source. LC separation was performed on an InfinityLab Poroshell 120 HILIC-Z, 2.1 mm × 50 mm, 2.7 µm UHPLC column (Agilent Technologies 689775−924) coupled to an InfinityLab Poroshell 120 HILIC-Z, 3.0 mm × 2.7 µm UHPLC guard column (Agilent Technologies 823750−948). A 3.5 min gradient was run using organic buffer (acetonitrile) combined with an aqueous buffer with low pH (10 mM ammonium formate, pH 3) or high pH (10 mM ammonium acetate, pH 9) for positive and negative ionisation modes, respectively. Data was acquired using MassHunter Data Acquisition 10.0 software on 1 µL of sample separated on the column with a flow rate of 800 µL min$^{-1}$. A pooled quality control sample was generated by combining equal volumes of each sample and injected five times at the beginning of the experiment to condition the column and after every five test samples to monitor the instrument state throughout data acquisition. Data were acquired in the 50 to 1700 m/z range, with an MS acquisition rate of 0.8 scans/s. The raw data files were processed using the Agilent MassHunter software suite. Briefly, ion multiplexed data files and calibration files were demultiplexed using the PNNL PreProcessor v2020.03.23 (the default settings for demultiplexing, moving average smoothing, saturation repair and spike removal were applied to the data). The data files were recalibrated for accurate mass and drift time using the AgtTofReprocessUi and the IM-MS Browser 10.0, respectively. Molecular features were extracted in Mass Profiler 10.0 with a retention time tolerance of ±0.3 min, drift time tolerance of ±1.5% and accurate mass tolerance of ±(5 ppm + 2 mDa). Features were annotated based on accurate mass and collision cross section values using a manually curated reference library consisting of amino acids and all possible combinations of dipeptides and tripeptides. The input data was sample normalised by protein concentration, log-transformed and Pareto-scaled. The results were visualised using variable importance in projection and heatmaps.

## Statistical analysis

All error bars represent a 95% confidence interval. Statistics and graphical analyses were performed using GraphPad Prism 10.4 (GraphPad Software, inc.). All data points represent independent experiments. All statistical metrics are provided in the figure legends. Multivariate statistical analysis for peptide analysis was performed using the MetaboAnalyst 6.0 web-based platform[39]. For some bacterial growth experiments, independent biological replicates were established, and growth was monitored simultaneously. Therefore, to ensure the robustness of the data, independent batched experiments where growth was monitored independently are provided in the Supplementary Data, where appropriate. The independent batched experiments used alternative live cell isolates and independent bacterial lysates to demonstrate the robustness of the observations.

## Reporting summary

Further information on research design is available in the Nature Portfolio Reporting Summary linked to this article.

## Data availability

Source data are provided with this paper. Peptide analysis data is availability through Metabolights study number MTBLS10044. Source data are provided with this paper.

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

## Acknowledgements

We thank Laurence Belcher for helpful discussion. We acknowledge European Research Council funding (834164) to S.A.W.

## Author contributions

S.E.R.G, I.F., S.H., and T.M. performed experiments and analysed data. S.E.R.G., T.M., S.A.W. and M.J.C. designed the experiments. W.C.K.P. and M.J.C. conceived the project. M.J.C. managed the project. M.J.C. and S.A.W. wrote the manuscript. All authors discussed the results and implications and commented on the manuscript at all stages.

## Competing interests

The authors declare no competing interests.
