## [Transparent Peer Review file · Nature Communications]

Bacteria encode post-mortem protein catabolism that enables altruistic nutrient recycling

Corresponding Author: Professor Martin Cann

Version 0:

Reviewer comments:

Reviewer #1

(Remarks to the Author)

I love the topic of this paper. I have long wondered if there might be kin selected benefits after death and this is the first study I have seen that addresses the question. So I score this study very high for novelty/originality.

That said, the paper is most exciting if the post-mortem effects demonstrated are true adaptations – that is if natural selection acts from beyond the grave in the wild. I think the authors believe this is true, or want us to believe that it is true, but they never quite come out and say it, perhaps because the issue is not simple. I would like to see the arguments pro and con. They do demonstrate that under lab conditions, there are the kinds of costs and benefits involved in a cooperation/cheating system. How do we know that lon protease effects after death are not simply side effects of lon protease benefit while alive? The cell has these proteases and then it dies and they happen to have additional consequences, but the post-death cooperation and cheating effects that occur in the lab might be selectively irrelevant in nature. It depends on the size of the cost, benefits, and especially relatedness. I should add that, even if this adaptation issue is uncertain, the paper is still interesting from a recycling mechanism standpoint. But it would be good to see the adaptation issue addressed head-on.

I found the paper rather difficult to read. I had to work hard to be sure of the logic behind some of the experiments. This could be improved with more signposting along the lines of “If our hypothesis X is true, then we expect the following differences...”. It might also help to occasionally refresh the reader’s memory of what the various strains are. And it would be useful if a way could be found to indicate significant differences on the figures themselves instead of having to wade through a lot of text in the figure captions.

32 By transport I assume you mean taking the peptide in from outside the cell?

52-55 Several claims are made here for which I did not find statistics. Did I miss it somehow?

64 “WT vs lon, BL21, and H2O $p < 0.0001$ ” Does this mean there are three comparisons with H2O or just one combined?

Fig. 2B Here you are testing whether each lysate is significantly different from background. But the real interest is in whether some lysates produce a higher signal than others. In all your other tests, you compare treatments against each other and I think that is what is best here as well.

Fig 2AB. I searched but could not find what +V stands for.

122-3 “WT vs lon, lon+ V, lon+ Lon S679A, lon+ Lon K362Q/S679A, H2O $p < 0.000$ ” Again, is this a single comparison with WT, or multiple comparisons? See also 170-1 and there may be other lists like this that I did not flag (I did not look closely at the supplemental figures for this).

257 paragraph. This looks to me to be an argument that cooperation is stabilized by a pleiotropic effect. That is curious because West has argued that this mechanism won’t work (Dos Santos et al. 2018 Plos Biology). Shouldn’t this be discussed?

Reviewer #2

(Remarks to the Author)

This study examines the effects and evolution of the *E. coli* Lon locus in the breakdown of proteins from dead cells to support the growth of surviving ones. The Lon locus and its action are topics of genetic and evolutionary interest. I found the abstract exciting, but found that the study, as conducted and presented, did not meet the criteria of being a study that represents an advance that influences "thinking in the field, with strong evidence for their conclusions." The conceptual integration, analysis of the study, and the presentation could be improved. The authors should be encouraged to develop further this study. The writing could be directed to a broader audience in microbiology or evolution, as opposed to a reader possibly already familiar with the study.

Lon is a fascinating system, but choices need to be made to develop a more integrated approach. Two possibilities are focusing on either the genetic or molecular basis of Lon or the evolution of Lon. This study has done both, but unfortunately left two key questions unanswered.

First, how did Lon evolve by kin selection? The frequency dependent experiments show the evolution of cheating, but that is not equivalent to the evolution of post-mortem Lon. The evolution of cooperation or altruism is not as big of a mystery as it is made out to be. Especially for microbes, kin selection is the preferred explanation (see e.g. work on social bacteria, slime molds, and programmed cell death). By falling back on kin selection, the study does not provide a solution to an evolutionary puzzle. A more significant contribution would be to demonstrate the evolution of Lon WT versus *delLon* by kin selection without sonication, which is an artifact points the difficulty of evolving post-mortem public goods.

Second, is Lon beneficial before death? The results from Line 139 paragraph show that Lon can act after death, but that does not exclude that it could be acting before.

Comments

Line 40 paragraph. Why was sonication and immediate death preferred over slower acting methods of inducing cell death? One would think that slower death would be closer to the form of death experienced by cells in conditions that lead to the evolution of the phenomenon.

OD measures biomass and CFU viable cells. Both can be viewed as a measure of fitness, but the authors need to pick one that they think best fits the requirements of their evolutionary perspectives. If it is important for them that OD and CFU are equivalent, the reasons why need to be stated and verified by more than that they were "indistinguishable". A statistical analysis would be needed to show that all the slopes are the same. However, because the slope regressions will have different error variances, all that need to be included in the final statistical analysis of the experimental data. I would suggest that they just use one (CFU or OD), which would greatly simplify the statistical analysis.

Line 48 paragraph. The authors show that *delLon* lysates do not enhance the growth of WT bacteria. The authors should give one or two sentences on the reason why *delLon* was chosen. Blind screening of hundreds of knockouts? Did previous studies hint or show enhancement, in which case they should be credited and described? The results with *delOmpT* are unnecessary. The study has the *delLon* knockout and that is sufficient, at this stage of the paper. If *delOmpT* had an important historical role, it should be described (see above). As presented, *delOmpT* only confuses the reader. The sentence starting on line 53 "We exclude the possibility.....", which is important, has no statistical validation. P-values are given in Figs. 1 and S2, but never for the *delLon* vs. water comparison. Additionally, Figs. 1 and S2 appear to be based on identical experiment, so why are the two not pooled (for a larger sample size), rather than making one supplemental? Visually inspecting Figs. 1 and S2, it is noticeable that water->*delLon*, and a larger sample size could show significance. Finally, it would have helped understanding the logic of line 53 if it were stated that it assumes that *delLon* makes the toxin. Despite a word limit, a hint here would be helpful (more than *delOmpT*).

Line 66 paragraph. This paragraph could be shortened or omitted because it does not contribute much to understanding the paper. In fact, it is confusing because the reader is led to believe that ATP is important, only to read later that Lon is active without ATP. If there are facts that are critical, they could be woven into the introduction, the presentation of results, or in the discussion.

Line 76 paragraph. It is not clear if this paragraph is summarizing completed experiments (that will not be describe in detail), or informing on experiments to be conducted. If it is the latter, I would suggest combining Line 76 and Line 84 into one paragraph.

Line 91 paragraph. It is not clear why this paragraph is needed because no new results are generated. As it is stated, it is already known that Lon is a protease that can use as a substrate Casein. If these experiments were controls conducted to validate the phenotype of the strain, they should be described either in the Material and Methods or Supplement. Line 105. "likely....within the error of the experiment." This should be validated with statistics, in the best case with a power test to show that sample size is sufficiently large.

Line 106 paragraph. The first sentence was very confusing. Lon is produced in live cells, and if the enhancement is real, what role is to be excluded? Is the question whether Lon is needed for a cell to benefit from enhancement? Otherwise, was not the role of Lon on enhancement already establish earlier? What is V? It appears for the first time in line 120 as complementing empty, which is not helpful. Is it a control? The figure legend needs to be rewritten so that the reader can

more readily compare that relevant phenotypes (rather than strain names). The words complemented or not-complemented are in Fig. 2C but never mentioned in this paragraph. That makes it difficult to connect the text to the figure because previous use of complementation involved plasmid and none were mentioned here. From the *B. subtilis* experiment it appears as if a goal was to show that enhancement acts cross species. If that was the goal, then why were the experiments on lines 106 to 110 needed? This paragraph was a difficult read. Reading between the lines 106 to 110, I thought that the question whether Lon is needed for a cell to benefit from enhancement. However, that does not fit the *B. subtilis* experiments, and it becomes less clear.

Line 139 paragraph. The sentence “we cannot exclude the possibility that the essential Lon protease-derived mechanism occurred in the cell before killing but only manifested post-mortem” is confusing because the breakdown of proteins by Lon is manifested post-mortem. The authors could clarify by providing a for example. In Line 142, a recombinant Lon is mentioned for the first time in this manuscript and it should be described in more detail and justified why it is needed (rather than just WT). The paragraph ends by concluding “Lon protease’s function is post-mortem and not dependent on its production in the cell before death.” Without a for example, this conclusion is not justified. For example, could the breakdown of proteins by Lon have occurred in live cells before death? Their experiments show that Lon proteases can break proteins when added post-mortem, but they do not exclude that protein breakdown by Lon evolved to act before death. To make it even more complicated, consider that Lon evolved to breakdown protein in live cells as part of a repair mechanism in stressed cells, possibly at the edge of death. However, Lon is in sufficiently high to provide protease function after cell death. Thus, the post-mortem effect is not an adaptation evolved through natural selection but rather an indirect side effect (Lewontin and Gould, 1979). There may be a key assumption or point that I am missing, but my interpretation is that showing that post-mortem is possible does not exclude in vivo function or adaptation.

Line 154 paragraph. This paragraph should be combined with line 106 paragraph. However, a point to be addressed is how does this paragraph help us understand the evolution of Lon or the mechanism of Lon?

Line 176 paragraph. Move to discussion?

Line 186 paragraph. How does this help us understand the post-mortem function of Lon, or its evolution?

Line 199 paragraph. If this is a view, then should figure a way to weave it with Line 19 paragraph. However, why does the abstract suggest kin selection? The manuscript needs to be better integrated conceptually. This paragraph could be moved to the discussion.

Line 206 paragraph. Kin selection is an alternative to the “side effect” evolution of Lon. However, bringing up kin selection points to the problem that this study does not help us understand better the evolution of Lon, despite the abstract claiming that the authors have a solution to the evolutionary puzzle of post-mortem phenotype. Perhaps move this paragraph to the discussion.

Line 211 paragraph. Good result. There is a cost that Lon cells pay that is manifested with low stress. At higher level of stress, WT and delLon did equally well. Line 217: Note that the cost is not absent. It is likely that there is now benefit that offsets the cost for Lon cells.

Line 234 paragraph. Merge with line 206 paragraph and place in discussion?

Line 257 paragraph. What are the hallmarks of kin selection? If it is reference to the frequency dependence, frequency dependence as presented is more of a hallmark of cheaters and not kin selection. Even then it is a weak hallmark because many other processes could lead to negative frequency dependence. The reference to Fig. 5D as stress free is confusing because it was previously used as a case of increasing stress (line 214 where 5 days stationary leads to significant death relative to early stationary). Additionally, line 232 mentions 5 days but the Fig. 5D legend does not. The authors should check for other similar inconsistencies because that may have led to my misreading and misinterpreting this manuscript.

Lines 211 to 263. The authors may want to consider that all of the experiments contained in these lines could be unified or combined thematically as represent what happens to del Lon and WT when stress and death rates are increased by exposure to longer stationary and heat shock. At low levels (early stationary) delLon has higher fitness because Lon has a cost. As stress/death is increased, delLon and WT transitions from having equal fitness (cost is balanced by benefits) to finally when WT wins. This presents an integrated overview that may be useful to a reader.

Line 264 paragraph. Extending this study to ecosystems enters a dangerous zone. While there are biologists that believe in ecosystems evolution (Gaia?), I would say that the majority evolutionary ecologists consider it discredited except in cases of mutualism in which the participants are joined almost permanently (e.g. mitochondria and a eukaryotic cell). One could image Lon like system in mitochondria or other organelles in a eukaryote. The permanency replaces kinship. However, details matter and the generality of this last paragraph weakens the manuscript by highlighting the need to be focusing more on specific hypotheses, such as is Lon benefiting before death or kin selection.

Version 2:

Reviewer comments:

Reviewer #1

(Remarks to the Author)

This revision satisfies my concerns with the first submission. I think this is the best evidence yet for kin-selected post-mortem effects and the revision provides better evidence for it being an adaptation in its own right rather than a side-effect of another adaptation. As such, it provides a nice expansion of the domain of kin selection.

Reviewer #3

(Remarks to the Author)

I find this a very interesting study with many excellent experiments and intriguing implications, but have concerns about some stated inferences. The study nicely establishes the plausibility of the hypothesis that post-mortem nutrient cycling mediated by Lon protease evolved a social adaptation in natural populations of bacteria, but it does not exclude the evolutionary byproduct hypothesis. Establishing such plausibility is important, even short of conclusively demonstrating an adaptation hypothesis.

Most of the authors' responses to Reviewer 2 seem very good, but I'm not satisfied with some. My biggest concern is that claims are stronger than warranted in several places, both in the manuscript and in the response to Reviewer 2. Yet some of the relevant phrasing is appropriately cautious, sometimes making for a confusing mix of boldly stated claims and softened suggestions on a given point (see example below). Inferences regarding past natural adaptation should be consistently cautious due to strong limitations on what can be strongly inferred from experiments performed under a very limited set of simple laboratory ecological conditions.

The authors nicely quantified relevant costs and benefits under a limited set of lab conditions, which is great, but they sometimes appear to draw too much from this. My key point is that ecological conditions can greatly affect the costs and benefits of both private and social traits. Thus, it can't be clearly inferred that the private benefits of Lon production across the broad range of natural environments in which Lon production originally evolved are insufficient to account for its evolution and that kin selection must be invoked.

To be clear, I think the social adaptation / kin selection hypothesis may very well be correct, and I consider this study a major first step in exploring this hypothesis. But I think the authors should be content with establishing plausibility — a significant accomplishment in its own right — and not try to claim more than that.

A few comments on the response document:

p. 5: "First, we show that a post-mortem effect is an adaptation. Second, we show that this post-mortem effect is an exploitable social adaptation." I disagree that these claims were demonstrated. What was shown is a post-mortem benefit of Lon under a very limited set of lab conditions. This is very interesting but not the same as convincingly showing that such an effect evolved as a social adaptation under natural conditions.

p. 6: "However, and crucially, this benefit of Lon to the live cell does not outweigh the cost (so net cost at private level)." Again, this statement only applies to the limited conditions examined. This may not be the case overall for the relevant range of historical natural environments. Private benefits might still explain Lon evolution, and I think the authors should state this.

p. 15: Based on the same logic as above, the byproduct hypothesis can't be excluded. That the byproduct hypothesis still remains plausible in light of the authors' results should be stated.

On a different point, the authors in one place argue that sonication is superior because it removes potential confounding factors during natural cell death, such as increased Lon production from stress responses, but then for new experiments in fact examine natural cell death. What do the authors' concerns about complications from natural cell death impact interpretation of the stationary-phase experiments? T

Comments on manuscript (with line # references):

The title phrasing seems too adaptationist, as the use of "to" suggests that adaptation for post-mortem nutrient recycling has been demonstrated, which I think is not the case. More neutral would be: "Bacteria encode post-mortem protein catabolism that enables nutrient recycling"

17: "We also showed that the production of Lon protease cannot be explained by a personal benefit to living cells." This seems to be stated as a general claim, but it can't be inferred generally because of the potential for ecological specificity. Thus, I don't think that has been shown.

19: "this private benefit does not outweigh the cost." This is only known for the conditions tested.

22: I think a better phrasing would be "This finding of an unexpected post-mortem biochemistry may fundamentally revise our understanding of the nutrient recycling process."

69: Here the use of "suggests" is correct.

314: "Under certain conditions..." Here the authors implicitly acknowledge out that ecological conditions can impact costs

and benefits.

337: "Private benefit does not outweigh cost." The authors should state that this conclusion only applies to their tested conditions. Also for line 345.

346: Here the language is correctly cautious..."appears to be" and "likely"

As an example of inconsistent phrasing, in line 327 it is written that "Lon protease is a post-mortem social adaptaion" whereas in line 346 the phrasing is appropriately more cautious: "appears to be a social adaptation". Line 327 should be modified.

344-45: I think better phrasing here would be "this private benefit did not outweigh the cost under the conditions examined."

356: Good cautious phrasing.

Version 3:

Reviewer comments:

Reviewer #3

(Remarks to the Author)

The authors have effectively addressed my comments and I have no further changes to suggest.
Very nice paper!

We thank the reviewers for the positive and constructive reviews. Our response is provided below.

The response is indicated in blue text. Amended text, where appropriate to include here, is shown in black (unchanged) and red text (changed).

As requested, we have expanded our work to:

(a) show that the post-mortem effect of Lon is a true adaptation and not just a byproduct benefit;

(b) clarified the molecular basis of Lon;

(c) address the unanswered questions.

Our most substantive new inclusions include:

1. We add additional data as Figures 5F-G to demonstrate that the benefit of Lon in living cells is a private trait that cannot be exploited by cheats (in contrast to the post-mortem effect).
2. New data in Figures 4A-D to provide further information on the molecular function of Lon and demonstrate that its function is post-mortem.
3. A new statistical analysis of the CFU vs OD curves (Figure S1)
4. Additional statistical information, where requested
5. A significant rewrite of large parts of the manuscript and complete re-write of the evolutionary information to provide clarity. In particular, we have clarified direct versus indirect adaptation.
6. Updated figures.

REVIEWER COMMENTS

Reviewer #1 (Remarks to the Author):

I love the topic of this paper. I have long wondered if there might be kin selected benefits after death and this is the first study I have seen that addresses the question. So I score this study very high for novelty/originality.

That said, the paper is most exciting if the post-mortem effects demonstrated are true adaptations – that is if natural selection acts from beyond the grave in the wild. I think the authors believe this is true, or want us to believe that it is true, but they never quite come out and say it, perhaps because the issue is not simple. I would like to see the arguments pro and con. They do demonstrate that under lab conditions, there are the kinds of costs and benefits involved in a cooperation/cheating system. How do we know

that Lon protease effects after death are not simply side effects of Lon protease benefit while alive? The cell has these proteases and then it dies and they happen to have additional consequences, but the post-death cooperation and cheating effects that occur in the lab might be selectively irrelevant in nature. It depends on the size of the cost, benefits, and especially relatedness. I should add that, even if this adaptation issue is uncertain, the paper is still interesting from a recycling mechanism standpoint. But it would be good to see the adaptation issue addressed head-on.

We agree completely re true adaptations versus byproduct and apologise for the confusion here. We have added extra data (Fig. 5F-G) and completely re-written the final section of the paper to tackle the adaptation issue more head-on. Our results show that even under stressful conditions, Lon protease production is a costly cooperative trait. This means, that in order to be favoured by natural selection it requires kin selection on the post-mortem effect. The post-mortem effect is therefore an adaptation. We have also rewritten the relevant parts of the summary. As we have clarified the issue of adaptation in the text, we propose no further clarifying experiments are required on this specific issue.

I found the paper rather difficult to read. I had to work hard to be sure of the logic behind some of the experiments. This could be improved with more signposting along the lines of "If our hypothesis X is true, then we expect the following differences...". It might also help to occasionally refresh the reader's memory of what the various strains are. And it would be useful if a way could be found to indicate significant differences on the figures themselves instead of having to wade through a lot of text in the figure captions.

We have added significant differences onto all figures throughout the manuscript. We have added those p-values most critical to interpret the figure.

We have also added subheadings to better structure the different hypotheses. We have further made efforts to reduced jargon by providing clear shorthand for strain names throughout the manuscript (to refresh function for the reader). We have made significant additions to the text to improve readability throughout.

32 By transport I assume you mean taking the peptide in from outside the cell?

Yes, we have amended the text to read (lines 31-32)

"Bacteria typically transport small peptides **into the cell** to serve as a source of fixed nitrogen and amino acids"

52-55 Several claims are made here for which I did not find statistics. Did I miss it somehow?

Yes, the p values are provided in the Figure legend for Figure 1. However, in response to the comment above, we have also made significant differences clearer on figures.

64 “WT vs Δlon , BL21, and H₂O $p < 0.0001$ ” Does this mean there are three comparisons with H₂O or just one combined?

Yes, there are three comparisons and we have amended the Figure 1 legend to make this more clear

“WT vs Δlon , WT vs BL21, and WT vs H₂O $p < 0.0001$; WT vs *DompT* $p = 0.9788$.”

Fig. 2B Here you are testing whether each lysate is significantly different from background. But the real interest is in whether some lysates produce a higher signal than others. In all your other tests, you compare treatments against each other and I think that is what is best here as well.

Yes, this is a good point. We have made the appropriate change to the Figure legend and updated with the requested comparisons which do not alter the findings.

Fig 2AB. I searched but could not find what +V stands for.

We apologise for this omission. ‘V’ is now clarified in the methodology in the Protein Production section (lines 483-484).

“All complementation analyses used the same pBAD33 constructs carrying the indicated *lon* variants. The empty pBAD33 plasmid is denoted as ‘V’ in the text.”

The information is also provided in the Figure legend for Figure 2.

122-3 “WT vs Δlon , $\Delta lon + V$, $\Delta lon + Lon S679A$, $\Delta lon + Lon K362Q/S679A$, H₂O $p < 0.000$ ” Again, is this a single comparison with WT, or multiple comparisons? See also 170-1 and there may be other lists like this that I did not flag (I did not look closely at the supplemental figures for this).

We apologise for this lack of clarity. We have gone through the manuscript in full and expanded all comparisons to remove any ambiguity.

257 paragraph. This looks to me to be an argument that cooperation is stabilized by a pleiotropic effect. That is curious because West has argued that this mechanism won’t work (Dos Santos et al. 2018 Plos Biology). Shouldn’t this be discussed?

There are two issues here. (1) We are not making a pleiotropy argument here as the private benefit does not outweigh the cost - we have now made this clearer in the main text. (2) Dos Santos et al. showed that pleiotropy can stabilise cooperation - but only if the private and cooperative traits must be linked (i.e. genetic architecture cannot evolve to split them). We have no idea if this is the case here, but we also do not need to make this assumption (see 1).

Reviewer #2 (Remarks to the Author):

This study examines the effects and evolution of the E. coli Lon locus in the breakdown of proteins from dead cells to support the growth of surviving ones. The Lon locus and its action are topics of genetic and evolutionary interest. I found the abstract exciting, but found that the study, as conducted and presented, did not meet the criteria of being a study that represents an advance that influences “thinking in the field, with strong evidence for their conclusions.” The conceptual integration, analysis of the study, and the presentation could be improved. The authors should be encouraged to develop further this study. The writing could be directed to a broader audience in microbiology or evolution, as opposed to a reader possibly already familiar with the study.

We thank the reviewer for their comments. We note here (detailed fully below) that we have performed a significant re-write of large parts of the manuscript for clarity and accessibility. Further, we have added new data to develop and clarify our findings. As detailed below, there is significant new data on the molecular function of Lon (Figure 4A-D) and the fitness consequences in living cells (Figure 5F-G).

Lon is a fascinating system, but choices need to be made to develop a more integrated approach. Two possibilities are focusing on either the genetic or molecular basis of Lon or the evolution of Lon. This study has done both, but unfortunately left two key questions unanswered.

First, how did Lon evolve by kin selection? The frequency dependent experiments show the evolution of cheating, but that is not equivalent to the evolution of post-mortem Lon. The evolution of cooperation or altruism is not as big of a mystery as it is made out to be. Especially for microbes, kin selection is the preferred explanation (see e.g. work on social bacteria, slime molds, and programmed cell death). By falling back on kin selection, the study does not provide a solution to an evolutionary puzzle.

We agree very strongly that kin selection can explain the evolution of costly altruistic traits in bacteria. Our aim here was to investigate the social costs and benefits of Lon to determine whether it actually is a costly altruistic trait. We found that Lon protease: (a) is costly to produce, but provides a public benefit; (b) can also provide a private benefit under stressful conditions, but that this does not

outweigh the cost. Therefore, it is a costly altruistic trait, likely to be explained by kin selection.

We have rewritten the final section of the paper to make the social costs and benefits clearer. In particular, see the "Lon protease is a social adaptation" section (Lines 279-353).

We have divided this section up to show how our data addresses: (i) Lon protease production is costly and exploitable; (ii) Public goods and kin selection; (iii) Lon protease can provide a private benefit; (iv) Private benefit does not outweigh cost. We have also rewritten the relevant parts of the summary.

As part of this re-write we have included additional data in Figures 5F-G to demonstrate that the benefit of Lon in living cells is a private trait that cannot be exploited by cheats (i.e. not cooperative - in contrast to the post-mortem effect).

We agree that all adaptations must be explained by direct or indirect (kin selected) benefits. The novelty of our work is two-fold. First, we show that a post-mortem effect is an exploitable social adaptation. Second, we show that this post-mortem effect is an exploitable social adaptation. We agree that this must be explained by kin selection, and that kin selection is widely accepted to explain many cooperative traits (across the tree of life). We hope that our new data and text rewrite has clarified this issue.

A more significant contribution would be to demonstrate the evolution of Lon WT versus delLon by kin selection without sonication, which is an artifact points the difficulty of evolving post-mortem public goods.

Our understanding of this comment is that the referee is referring to the data presented in Figure 5 which deals with kin selection. None of the experiments presented here have used sonication but rather rely on natural cell death in stationary phase. We propose that no additional experiment is required here as sonication is not a feature of this part of the study. We have demonstrated that Lon is a costly trait, requiring a post-mortem benefit to be selected for, under conditions of normal cell death.

We have clarified this in the text (lines 297-298). More generally, as discussed above, we have rewritten our coverage of issue in the "Lon protease is a social adaptation" section (Lines 279-353). We have also clarified the relevant parts of the summary.

“Importantly, this experiment was performed with natural cell death at stationary phase, not induced cell death by sonication.”

Second, is Lon beneficial before death? The results from Line 139 paragraph show that Lon can act after death, but that does not exclude that it could be acting before.

Yes, Lon can also provide a benefit before death. First, this observation is well known from the literature and we have added a reference to evidence this and have amended the text (line 64-68).

“An *E. coli* BW25113 $\Delta ompT$ lysate (*OmpT*-null) enhanced *E. coli* growth compared to the H₂O control, but an *E. coli* BW25113 Δlon lysate (*Lon*-null) did not. Lon protease has a beneficial role in stress responses in live bacteria ⁸. It is an energy-dependent AAA+ (ATPases associated with cellular activities) protease that performs intracellular ATP-regulated protein degradation to recognise, unfold, translocate, and degrade substrates ⁹.”

Second, this finding is evident from our data. For example, Fig 5D demonstrates that the Lon protease is required for log phase growth under a stress condition (elevated temperature).

Therefore, we propose that no additional work is required here as the reviewer comment is addressed by the literature and data within the manuscript.

We have made this clearer (lines 322-323).

“This suggests that Lon protease production can provide a benefit to the live cell other than the post-mortem public benefit.”

However, and crucially, this benefit of Lon to the live cell does not outweigh the cost (so net cost at private level). We have re-written the parts of the manuscript dealing with evolution. We write specifically on the public versus private benefits of Lon and how they can co-exist. See the subsection “*Private benefit does not outweigh cost*”. We have also rewritten the relevant parts of the summary.

Comments

Line 40 paragraph. Why was sonication and immediate death preferred over slowing acting methods of inducing cell death? One would think that slower death would be closer to the form of death experienced by cells in conditions that lead to the evolution of the phenomenon.

We selected sonication for the purposes of this study as death is (almost) instantaneous. Therefore, our methodology avoids potential adaptive responses by bacteria that might be induced under the stress of a slower death. Such adaptive responses include up regulation of Lon protein production. Thus, we avoid ambiguity in the interpretation of our findings by removing such stress responses. This rationale is made in the second paragraph and we expand to make more clear (lines 47-51).

"Lysis via sonication was used as it was immediate and did not permit physiological changes in the bacteria that might be induced by other, slower-acting methods of inducing cell death. Such physiological responses, while normal for a bacterial response to stress, might lead to ambiguity in the interpretation of findings and so are avoided in this study. The sterile lysates derived from killed bacteria were examined for their growth enhancement of live cells..."

OD measures biomass and CFU viable cells. Both can be viewed as a measure of fitness, but the authors need to pick one that they think best fits the requirements of their evolutionary perspectives. If it is important for them that OD and CFU are equivalent, the reasons why need to be stated and verified by more than that they were "indistinguishable". A statistical analysis would be needed to show that all the slopes are the same. However, because the slope regressions will have different error variances, all that need to be included in the final statistical analysis of the experimental data. I would suggest that they just use one (CFU or OD), which would greatly simplify the statistical analysis.

For Figure 1-3 we have used OD as a measure of biomass increase. The sheer volume of samples makes CFU measurements impractical for these experiments. Therefore, we agree with the referee that we should perform an analysis to demonstrate that the relevant OD/CFU regressions are statistically similar. We have included this new analysis with the legend for Figure S1 so as not to hinder the flow of the main text.

"The slopes for panels **A-D** are not significantly different ($p = 0.3577$; F (DFn = 3, DFd = 88) = 1.09). This means that changes in cell density for *E. coli* BW25113 cells is correlated with an equivalent change in cell number regardless of which bacterial lysate is used in the experiment.

The slopes for panels **E-G** are not significantly different ($p = 0.8582$; F (DFn = 2, DFd = 90) = 0.1532). This means that changes in cell density for *E. coli* BW25113 Δ Lon cells is correlated with an equivalent change in cell number regardless of which bacterial lysate is used in the experiment.

The slopes for panels **H-K** are not significantly different ($p = 0.4548$; F (DFn = 3, DFd = 52) = 0.8854). This means that changes in

cell density for *B. subtilis* cells is correlated with an equivalent change in cell number regardless of which bacterial lysate is used in the experiment.”

For Figures 1-3 there is no requirement to include the slope variances in the data analysis as we are not reporting any CFU mL⁻¹ values, nor are we using the curves of Figure S1 for the data reported in Figure 1-3. We agree with the referee that we need to be very explicit on this fact and so have amended the main text thus (lines 52-56)

“We confirmed, through a statistical analysis of observed regression slopes, that different bacterial lysates had no impact on calibration curves of D_{600} against colony-forming units (CFU) (Figure S1) for the different live bacterial strains used in this study. Therefore, while this study is measuring bacterial biomass (D_{600}) and not reporting CFU mL⁻¹, we infer that the biomass increase is likely to represent a real increase in cell number.”

For Figure 5, there are experiments performed over extended time frames (all panels except A and E). These experiments require us to use CFU/mL readings because the cell culture debris after extended cultures makes D_{600} unreliable. Here also, we do not conflate cell number with biomass but agree with the referee that the rationale for using CFU/mL in these experiments, while methodologically sound, needs justification. The data of Figure 5 is measured CFU/mL and not calculated from the data of Figure S1. This clarification has been made in the methodology (Fitness Assays Section) (lines 495-498)

“We used measured CFU counts for experiments that required cultured over five days, rather than D_{600} measurements, as dead cell debris rendered D_{600} measurements unreliable as a measure of live cell biomass.”

Line 48 paragraph. The authors show that delLon lysates do not enhance the growth of WT bacteria. The authors should give one or two sentences on the reason why delLon was chosen. Blind screening of hundreds of knockouts? Did previous studies hint or show enhancement, in which case they should be credited and described?

A very good question. We hypothesised a role for the proteases ablated in BL21 cells as they are so active in protein turnover. This is now made clearer in the manuscript (lines 59-63).

“*E. coli* BL21 is a bacterial strain commonly used for recombinant protein expression as it is ablated for genes encoding the Lon and OmpT proteases that might otherwise increase recombinant protein turnover⁷. Therefore, we examined

whether it might also have a role in post-mortem protein turnover. An *E. coli* BL21 lysate did not enhance live WT cell growth.”

The results with delOmpT are unnecessary. The study has the delLon knockout and that is sufficient, at this stage of the paper. If delOmpT had an important historical role, it should be described (see above). As presented, delOmpT only confuses the reader.

We had not made the usefulness of this data sufficiently clear. The data with OmpT demonstrates that the observation is unlikely be due to reducing a threshold level of protease activity in cell lysates that can be reached by knocking out any protease. Thus, the experiment does demonstrate an important principle. We have made this point clearer in the text (lines 78-80).

“Further, the observations are not likely explained by a non-specific reduction in lysate protease activity as only ablating *lon*, and not *ompT*, impacts growth i.e., there is no apparent redundancy between these proteases.”

The sentence starting on line 53 “We exclude the possibility....”, which is important, has no statistical validation. P-values are given in Figs. 1 and S2, but never for the delLon vs. water comparison.

We apologise for omitting this important information. These comparisons are now provided in the figure legends, and we draw attention in the text as shown below (lines 71-76).

“We exclude the possibility that the Lon protease in the bacterial lysate degrades an otherwise toxic molecule that inhibits growth, as bacterial growth is indistinguishable between live cultures grown with H₂O and a Δlon lysate (see H₂O vs Δlon comparisons in Figures 1B, 2A, 2C, 3A, S3F-G, S5B-C). We note that there is a significant difference between H₂O and Δlon in Figure S2A. However, this result is likely a false positive given a confidence level of 0.05 and the large number of independent observations where the two samples are statistically indistinguishable.”

Additionally, Figs. 1 and S2 appear to be based on identical experiment, so why are the two not pooled (for a larger sample size), rather than making one supplemental?

The data points in Figs 1 and S2 are all independently performed experiments. There is an additional difference between Fig 1 and Fig S2 in that the two data sets use independent live cell isolates and independent bacterial lysates, in addition to be performed independently. We propose that this is the most transparent way to present the data as

it demonstrates repeatability when altering every experimental variable. This point is made in the methodology under Statistical Analysis. However, we agree that this point should be made more explicit and have, therefore, amended this text as shown below (lines 536-541).

“For some bacterial growth experiments, Independent biological replicates were established, and growth was monitored simultaneously. Therefore, to ensure the robustness of the data, independent batched experiments where growth was monitored independently are provided in the Supplementary Data, where appropriate. The independent batched experiments used alternative live cell isolates and independent bacterial lysates to demonstrate the robustness of the observations.”

Visually inspecting Figs. 1 and S2, it is noticeable that water>delLon, and a larger sample size could show significance.

See comment above and amended text. At a confidence interval of 0.05 and the large number of data sets showing no difference, this is likely a false positive.

Finally, it would have helped understanding the logic of line 53 if it were stated that it assumes that delLon makes the toxin. Despite a word limit, a hint here would be helpful (more than delOmpT).

We have signposted the (now separate) paragraph that eliminates the possibility of a toxic molecule with the following leading sentence to provide further clarity (lines 70-71).

“It is possible that Lon-null lysates accumulate a toxic molecule that inhibits growth relative to wild type lysates.”

Line 66 paragraph. This paragraph could be shortened or omitted because it does not contribute much to understanding the paper. In fact, it is confusing because the reader is led to believe that ATP is important, only to read later that Lon is active without ATP. If there are facts that are critical, they could be woven into the introduction, the presentation of results, or in the discussion.

We have deleted this paragraph and redistributed some information in other parts of the manuscript where it provides the information as needed in context.

Line 76 paragraph. It is not clear if this paragraph is summarizing completed experiments (that will not be describe in detail), or informing on experiments to be conducted. If it is the latter, I would suggest combining Line 76 and Line 84 into one paragraph.

As we have made other changes to the text in this region of the manuscript, we have kept two paragraphs but made them clearer, as suggested.

Line 91 paragraph. It is not clear why this paragraph is needed because no new results are generated. As it is stated, it is already known that Lon is a protease that can use as a substrate Casein. If these experiments were controls conducted to validate the phenotype of the strain, they should be described either in the Material and Methods or Supplement. Line 105. "likely...within the error of the experiment." This should be validated with statistics, in the best case with a power test to show that sample size is sufficiently large.

We respectfully disagree with the referee that this paragraph provides no new results. Indeed, it is known that Lon is a protease. However, we demonstrate that the ability of Lon to provide a post-mortem growth enhancement directly correlates with measurable protease activity. We have made this clearer by providing a clarifying sentence at the close of the paragraph (lines 131-133).

"Therefore, the ability of Lon protease to provide a post-mortem growth enhancement is correlated with the measurable protease activity in the bacterial lysate."

Further, we propose that this experiment is important to contextualise the metabolomics data of Figure 4. Therefore, we have added further clarifying information later in the manuscript (lines 223-226).

"A role for Lon protease in enabling post-mortem nutrient utilisation and the observation that this phenotype correlates with the protease activity present in the lysates of killed bacteria, indicates that Lon specifically catabolises high molecular weight intact proteins to smaller molecular weight peptides for transport into the live bacteria."

We have performed a power analysis as requested by the referee which reveals that the sample size is sufficient to observe an effect. Therefore, we have updated the text as requested (lines 127-131).

"Although the observation of no significant measurable protease activity with lysates derived from Δlon cells complemented with plasmid producing Lon-K362Q would seem to conflict with the finding that the Lon-K362Q lysate can rescue the Δlon lysate phenotype, it is most likely that activity in this lysate is within the error of the experiment as a power calculation shows that the sample size is sufficient to observe a difference."

Line 106 paragraph. The first sentence was very confusing. Lon is produced in live cells, and if the enhancement is real, what role is to be excluded? Is the question whether Lon is needed for a cell to benefit from enhancement? Otherwise, was not the role of Lon on enhancement already established earlier?

We apologise for this lack of clarity and see how this is unclear. The live bacterial cells (i.e., the cells whose growth is being enhanced) up to this point in the manuscript have been wild type cells. We wanted to eliminate the possibility of a role for the Lon protease in these cells. For example, it was possible that the observed phenotype was dependent on Lon protease released from dead cells but also that secreted by Lon cells. This experiment eliminated that possibility. We clarify this point through the following addition to this paragraph (lines 133-149)

"It is formally possible that Lon protease is both released from the dead cell (and, thus, present in the bacterial lysate) and secreted by the live cells (whose growth is being enhanced) to produce the observed phenotype. To exclude any role for Lon protease produced in the live cells used in the experiments, we performed additional experiments using Lon-null cells as the live culture (Figure 2C, Figure S4A-B.)"

What is V? It appears for the first time in line 120 as complementing empty, which is not helpful. Is it a control?

We apologise for this omission. 'V' is now clarified in the methodology in the Protein Production section (lines 483-484).

"All complementation analyses used the same pBAD33 constructs carrying the indicated lon variants. The empty pBAD33 plasmid is denoted as 'V' in the text."

The information is also provided in the Figure legend for Figure 2.

The figure legend needs to be rewritten so that the reader can more readily compare that relevant phenotypes (rather than strain names).

We have updated the figure legend with information on which comparisons demonstrate the complementation phenotype and which comparisons demonstrate that there is no complementation.

The words complemented or not-complemented are in Fig. 2C but never mentioned in this paragraph. That makes it difficult to connect the text to the figure because previous use of complementation involved plasmid and none were mentioned here.

While complementation is referred to in the text (we assume this is what the referee is referring to), we appreciate that it is not defined to guide the reader. Therefore, at first mention we define it thus (lines 91-92)

“Complementation restores a normal phenotype to mutants with an observable defect.”

From the *B. subtilis* experiment it appears as if a goal was to show that enhancement acts cross species. If that was the goal, then why were the experiments on lines 106 to 110 needed? This paragraph was a difficult read. Reading between the lines 106 to 110, I thought that the question whether Lon is needed for a cell to benefit from enhancement. However, that does not fit the *B. subtilis* experiments, and it becomes less clear.

Please see comment above that provides clarification for the 106-110 experiment.

We agree with the referee that the *B. subtilis* experiment following straight on is unclear and it gives the impression of a link between the two experiments which are intended to demonstrate independent ideas. Therefore, we have delineated this into two paragraphs to provide a natural break between the two experiments.

Line 139 paragraph. The sentence “we cannot exclude the possibility that the essential Lon protease-derived mechanism occurred in the cell before killing but only manifested post-mortem” is confusing because the breakdown of proteins by Lon is manifested post-mortem. The authors could clarify by providing a for example.

We have clarified this sentence and provided an example, as requested (lines 177-179).

“However, we cannot exclude the possibility that Lon protease functions in the live cell. For example, it is possible that Lon protease produces nutrients in the live cell that have a post-mortem function after release from dead cells.”

In Line 142, a recombinant Lon is mentioned for the first time in this manuscript and it should be described in more detail and justified why it is needed (rather than just WT).

We have clarified the use of recombinant Lon by adding the following explanatory material (lines 179-181).

“A purified recombinant Lon protease can be used to add Lon activity to the lysates of killed Δlon cells. This approach enables us to separate the post-mortem Lon function from the function of Lon produced in the live cell.”

The paragraph ends by concluding “Lon protease’s function is post-mortem and not dependent on its production in the cell before death.” Without a for example, this conclusion is not justified. For example, could the breakdown of proteins by Lon have occurred in live cells before death? Their experiments show that Lon proteases can break proteins when added post-mortem, but they do not exclude that protein breakdown by Lon evolved to act before death.

This is an interesting line of discussion from the referee and we thank the referee for prompting a new line of investigation. From the initial manuscript, we think a strong line of evidence is that the addition of Lon protease at levels consistent with the amount in the live cell (Figure S3) complements the Δlon phenotype fully without having to invoke the requirement for a role for Lon in the live cell (Figure 3A). Therefore, as extracellular Lon is sufficient to complement the Δlon phenotype, this excludes the possibility of higher levels of peptide due to Lon in the live organism.

However, in the absence of a method to directly observe peptides levels inside the cell, we have performed additional experiments to measure peptides immediately on release from cells and after an incubation consistent with post-mortem growth.

We first use a chemical method to measure peptide levels immediately after release from the cell. We observe no difference between WT and Lon-null cells (see further details in amended text (new Figure 4A)). This chemical method measures peptides in the 6-10 amino acid range. We then observe that amino acids in the 6-10 amino acid range are reduced after incubation post-mortem in WT but not Lon-null lysates (new Figure 4B). This is presumably due to catabolism to smaller peptides that can act as nutrients. To demonstrate this, we use an alternative mass spectrometry-based method and quantify the distributions of 923 amino acids, di- and tripeptides. We observe no difference in the distribution between WT and Lon-null cells for lysates immediately on release from cells (new Figure 4C). We then observe that the distribution of amino acids, di- and tri-peptides after incubation post-mortem increases in WT compared to Lon-null lysates (new Figure 4D).

To summarise, we make the following observations.

1. 6-10 amino acids peptide content in WT and Lon-null cells is indistinguishable directly after cell death (presumably reflecting in vivo content).

2. 6-10 amino acids peptide content decreases in WT but not Lon-null lysates after incubation, presumably being catabolised to smaller peptides.
3. Amino acid, di- and tri-peptide content in WT and Lon-null cells is indistinguishable directly after cell death (presumably reflecting in vivo content).
4. Amino acid, di- and tri-peptide content increases in WT compared Lon-null lysates after incubation

These observations are consistent with the measurements of di and tri-peptides from the initial manuscript. Thus, we have created a new narrative that describes the fate of peptides post-mortem.

However, the referee is correct that it is impossible to completely rule out differences in some peptides in the live cell (due to technical limitations) that contribute to nutrient recycling i.e., an additional role for Lon in producing peptides in the live cell (although we would argue that the data of Figure 3A makes this unlikely). Nonetheless, we have moderated the conclusion of the new text to reflect this degree of uncertainty.

The new text is too lengthy to include here but can be found as lines 232-265 of the new manuscript.

To make it even more complicated, consider that Lon evolved to breakdown protein in live cells as part of a repair mechanism in stressed cells, possibly at the edge of death. However, Lon is in sufficiently high to provide protease function after cell death. Thus, the post-mortem effect is not an adaptation evolved through natural selection but rather an indirect side effect (Lewontin and Gould, 1979). There may be a key assumption or point that I am missing, but my interpretation is that showing that post-mortem is possible does not exclude in vivo function or adaptation.

The referee raises the key issue that it is necessary to determine whether Lon could just be explained by its function in live cells, in which case the post-mortem effect could just be a byproduct benefit. We agree completely and apologise that we were not sufficiently clear on this issue. To address this, we have rewritten the relevant section of the paper, which is now titled: "Lon protease is a social adaptation" (Lines 279-353) and added new data (Fig. 5F-G).

In this section, we show that Lon protease can provide a benefit to live cells, but this benefit is outweighed by the cost. Consequently, the benefit to live cells cannot explain the maintenance of Lon protease production (i.e. it can't just be explained as a byproduct benefit). The selection for Lon protease production can only be explained when taking into account the post-mortem benefit. As we have clarified this

issue in the text, we propose that no additional experiment is required here.

Line 154 paragraph. This paragraph should be combined with line 106 paragraph. However, a point to be addressed is how does this paragraph help us understand the evolution of Lon or the mechanism of Lon?

We respectfully disagree with the referee that the line 154 paragraph should be combined with the line 106 paragraph. The reason is that this experiment follows directly from the previous paragraph using a recombinant Lon protease. We have added material at the beginning and end of this paragraph to justify the position of the paragraph and to draw attention to the general properties of Lon, in comparison to other proteases (lines 198-200 and 207-209).

"Recombinant Lon proteases can be used as a further tool to investigate compatibility between the growing bacterial species and the source of the Lon protease. Therefore we, investigated whether the Lon protease post-mortem function...

"Therefore, while a role in nutrient recycling is not a generic property of all proteases (e.g., see OmpT in Figure 1B), it is likely a generic property of all Lon proteases."

Line 176 paragraph. Move to discussion?

We favour keeping the material where it is, as it serves to contextualise the experiments for Figure 4. However, we agree that the amount of material is too much for this purpose. Therefore, rather than move it, we have reduced it, so it is easier to read and helps to understand the context of Figure 4. We have also added further contextualising information in the following paragraph (lines 223-234)

"A role for Lon protease in enabling post-mortem nutrient utilisation and the observation that this phenotype correlates with the protease activity present in the lysates of killed bacteria, indicates that Lon specifically catabolises high molecular weight intact proteins to smaller molecular weight peptides for transport into the live bacteria. Bacteria possess numerous peptide transporters for importing peptides as nutrients. There are two major categories of peptide transporters. First, the proton motive force-driven transporters (POT or PTR family), which cluster in pairs of YdgR/YhiP and YjdL/YbgH¹⁸ and utilise proton import for transport. Second, the ATP binding cassette-containing transporters (ABC transporters) that include the dipeptide permeases (DdpABCDF and DppABCDF) and the oligopeptide

permease (OppABCDF) ¹⁹. The most common peptide substrates for uptake are di- and tripeptides.

Given the role of Lon protease in post-mortem nutrient recycling and its link to measured protease activity in lysates derived from dead bacteria, we hypothesised that Lon protease produces di- and tripeptides for uptake by peptide transporters.”

Line 186 paragraph. How does this help us understand the post-mortem function of Lon, or its evolution?

This paragraph demonstrates that Lon protease generates peptides of the appropriate size for uptake as nutrients. This observation provides additional validation for its role in nutrient recycling. We have made this point more explicit in the new manuscript with the inclusion of new text (lines 232-265) and Figures 4A-D.

Line 199 paragraph. If this is a view, then should figure a way to weave it with Line 19 paragraph. However, why does the abstract suggest kin selection? The manuscript needs to be better integrated conceptually. This paragraph could be moved to the discussion.

We apologise for lack of clarity here. We have completely rewritten the final part of the manuscript, to clarify the social and private nature of the trait. We have also clarified in the abstract which has been rewritten.

Line 206 paragraph. Kin selection is an alternative to the “side effect” evolution of Lon. However, bringing up kin selection points to the problem that this study does not help us understand better the evolution of Lon, despite the abstract claiming that the authors have a solution to the evolutionary puzzle of post-mortem phenotype. Perhaps move this paragraph to the discussion.

We have rewritten this section to separate and better clarify the different points, especially what we have and haven't shown. Our new version combines the relevant parts in a single section: “Lon protease is a social adaptation” (Lines 279-353). Within this, we show how the cost outweighs the private benefit, such that the “side effect” hypothesis can be rejected.

We have also rewritten the relevant parts of the abstract (summary) to clarify.

Line 211 paragraph. Good result. There is a cost that Lon cells pay that is manifested with low stress. At higher level of stress, WT and delLon did equally well. Line 217: Note

that the cost is not absent. It is likely that there is now benefit that offsets the cost for Lon cells.

We agree with the referee, and this is a good point. We have rewritten this section.

Line 234 paragraph. Merge with line 206 paragraph and place in discussion?

We have rewritten this section.

Line 257 paragraph. What are the hallmarks of kin selection? If it is reference to the frequency dependence, frequency dependence as presented is more of a hallmark of cheaters and not kin selection. Even then it is a weak hallmark because many other processes could lead to negative frequency dependence.

We have rewritten this section to clarify these points. The main purpose here was to show that the post-mortem function is an exploitable public good - i.e. that it is individually costly, and benefits others. The frequency dependence is just a little bonus on top of that - something else that is common with public goods, but only interesting in this case because have already shown private cost and group benefit (i.e. public good).

The reference to Fig. 5D as stress free is confusing because it was previously used as a case of increasing stress (line 214 where 5 days stationary leads to significant death relative to early stationary). Additionally, line 232 mentions 5 days but the Fig. 5D legend does not. The authors should check for other similar inconsistencies because that may have led to my misreading and misinterpreting this manuscript.

All inconsistencies have been addressed in the text and figure legend.

Lines 211 to 263. The authors may want to consider that all of the experiments contained in these lines could be unified or combined thematically as represent what happens to del Lon and WT when stress and death rates are increased by exposure to longer stationary and heat shock. At low levels (early stationary) delLon has higher fitness because Lon has a cost. As stress/death is increased, delLon and WT transitions from having equal fitness (cost is balanced by benefits) to finally when WT wins. This presents an integrated overview that may be useful to a reader.

We agree with the referee. We have tried to provide this overview in the new "Lon protease is a social adaptation" section.

Line 264 paragraph. Extending this study to ecosystems enters a dangerous zone. While there are biologists that believe in ecosystems evolution (Gaia?), I would say that the

majority evolutionary ecologists consider it discredited except in cases of mutualism in which the participants are joined almost permanently (e.g. mitochondria and a eukaryotic cell). One could image Lon like system in mitochondria or other organelles in a eukaryote. The permanency replaces kinship. However, details matter and the generality of this last paragraph weakens the manuscript by highlighting the need to be focusing more on specific hypotheses, such as is Lon benefiting before death or kin selection.

Apologies - we realise that our text was unclear. We were talking about post-mortem functions in other species, not between species. We have clarified this paragraph.

REVIEWER COMMENTS

Reviewer #1 (Remarks to the Author):

This revision satisfies my concerns with the first submission. I think this is the best evidence yet for kin-selected post-mortem effects and the revision provides better evidence for it being an adaptation in its own right rather than a side-effect of another adaptation. As such, it provides a nice expansion of the domain of kin selection.

We thank the reviewer. No further action required.

Reviewer #3 (Remarks to the Author):

I find this a very interesting study with many excellent experiments and intriguing implications, but have concerns about some stated inferences. The study nicely establishes the plausibility of the hypothesis that post-mortem nutrient cycling mediated by Lon protease evolved a social adaptation in natural populations of bacteria, but it does not exclude the evolutionary byproduct hypothesis. Establishing such plausibility is important, even short of conclusively demonstrating an adaptation hypothesis.

Most of the authors' responses to Reviewer 2 seem very good, but I'm not satisfied with some. My biggest concern is that claims are stronger than warranted in several places, both in the manuscript and in the response to Reviewer 2. Yet some of the relevant phrasing is appropriately cautious, sometimes making for a confusing mix of boldly stated claims and softened suggestions on a given point (see example below). Inferences regarding past natural adaptation should be consistently cautious due to strong limitations on what can be strongly inferred from experiments performed under a very limited set of simple laboratory ecological conditions.

The authors nicely quantified relevant costs and benefits under a limited set of lab conditions, which is great, but they sometimes appear to draw too much from this. My key point is that ecological conditions can greatly affect the costs and benefits of both private and social traits. Thus, it can't be clearly inferred that the private benefits of Lon production across the broad range of natural environments in which Lon production originally evolved are insufficient to account for its evolution and that kin selection must be invoked.

To be clear, I think the social adaptation / kin selection hypothesis may very well be correct, and I consider this study a major first step in exploring this hypothesis. But I think the authors should be content with establishing plausibility — a significant accomplishment in its own right — and not try to claim more than that.

We thank the reviewer for commenting on our response. Please see details of our response below.

A few comments on the response document:

p. 5: “First, we show that a post-mortem effect is an adaptation. Second, we show that this post-mortem effect is an exploitable social adaptation.” I disagree that these claims were demonstrated. What was shown is a post-mortem benefit of Lon under a very limited set of lab conditions. This is very interesting but not the same as convincingly showing that such an effect evolved as a social adaptation under natural conditions.

We have changed the text as suggested by the referee to make clear that our conclusions are for the conditions we examined: (i) in the abstract (line 20); (ii) subheading (line 279); (iii) main text (lines 338 & 348-355). Please also see changes in response to other comments below.

Line 20

“Although Lon protease can also provide a benefit to living cells under stressful conditions, by helping control protein quality, this private benefit does not outweigh the cost **under the conditions examined**”

Line 279

“Lon protease **appears to be** a post-mortem social adaptation”

Lines 349-356

“Nonetheless, taken together, our results suggest that Lon protease is a cooperative public good - a social adaptation that is likely to be partially explained by kin selection. Although it can also provide a private benefit under stressful conditions, we found that this private benefit does not outweigh the cost under our examined experimental conditions. This quantitative conclusion is for the conditions we examined, and the relative importance of private and public benefits could vary with environmental conditions, such as the relative occurrence of stressful conditions, and the potential gains from post-mortem nutrient utilisation. In the extreme, different scenarios could lead to the private benefit being either negligible or outweighing the cost.”

p. 6: “However, and crucially, this benefit of Lon to the live cell does not outweigh the cost (so net cost at private level).” Again, this statement only applies to the limited conditions examined. This may not be the case overall for the relevant range of historical natural environments. Private benefits might still explain Lon evolution, and I think the authors should state this.

See changes detailed above (main text lines 338 & 349-356).

p. 15: Based on the same logic as above, the byproduct hypothesis can't be excluded. That the byproduct hypothesis still remains plausible in light of the authors' results should be stated.

We have changed as suggested and detailed above - see especially lines 351-354. This is also made explicit in lines 339-340

“If this was the case, then the cooperative benefit of Lon protease could potentially be explained as a byproduct of a private trait, rather than a social adaptation.”

On a different point, the authors in one place argue that sonication is superior because it removes potential confounding factors during natural cell death, such as increased Lon production from stress responses, but then for new experiments in fact examine natural cell death. What do the authors' concerns about complications from natural cell death impact interpretation of the stationary-phase experiments? T

Yes, this is an interesting point. We acknowledge that the certainty of a lack of death-induced physiological changes is absent in the experiments of Figure 5. However, we note that the observations of the social adaptation studies are congruent with the more controlled experimental conditions of Figures 1-4. To address this, we have amended the text with a qualifying statement (lines 346-349)

“While the method of inducing cell death (sonication) in defining the post-mortem Lon function differs from the natural cell death occurring in these social adaptation studies, the findings are congruent. All the same, we cannot eliminate the possibility of a hitherto undefined physiological response in addition to Lon in natural cell death. Nonetheless, taken together, our results show that Lon protease is a cooperative public good.”

Comments on manuscript (with line # references):

The title phrasing seems too adaptationist, as the use of “to” suggests that adaptation for post-mortem nutrient recycling has been demonstrated, which I think is not the case. More neutral would be: "Bacteria encode post-mortem protein catabolism that enables nutrient recycling"

We have changed the title as requested. We have added the word “altruistically” as we show the effect is altruistic (under the tested conditions as clarified in the manuscript).

17: "We also showed that the production of Lon protease cannot be explained by a personal benefit to living cells." This seems to be stated as a general claim, but it can't be inferred generally because of the potential for ecological specificity. Thus, I don't think that has been shown.

See changes detailed above (response p5 comment).

19: "this private benefit does not outweigh the cost." This is only known for the conditions tested.

This sentence in the abstract has been amended to read (line 18-20) "Although Lon protease can also provide a benefit to living cells under stressful conditions, by helping control protein quality, this private benefit does not outweigh the cost **under the conditions examined.**"

22: I think a better phrasing would be "This finding of an unexpected post-mortem biochemistry may fundamentally revise our understanding of the nutrient recycling process."

Yes, we agree this sentence is better and have changed as requested (line 22-23).

69: Here the use of "suggests" is correct.

No action required.

314: "Under certain conditions..." Here the authors implicitly acknowledge out that ecological conditions can impact costs and benefits.

No action required.

337: "Private benefit does not outweigh cost." The authors should state that this conclusion only applies to their tested conditions. Also for line 345.

We have changed the respective sentences to read (lines 337-339)

"We then asked whether the private benefit of producing Lon protease can completely outweigh the cost **under our examined experimental conditions**, such that the trait provides a net benefit at the private level."

And (lines 350-352)

"Although it can also provide a private benefit under stressful conditions, this private benefit does not outweigh the cost **under our examined experimental conditions.**"

346: Here the language is correctly cautious..."appears to be" and "likely"

No action required

As an example of inconsistent phrasing, in line 327 it is written that "Lon protease is a post-mortem social adaptaion" whereas in line 346 the phrasing is appropriately more cautious: "appears to be a social adaptation". Line 327 should be modified.

We presume the reviewer is referring to Line 279 as this phrase does not appear at Line 327. We have altered line 279 to read

"Lon protease **appears to be** a post-mortem social adaptation"

344-45: I think better phrasing here would be "this private benefit did not outweigh the cost under the conditions examined."

The sentence has been altered to read (lines 350-352)

"Although it can also provide a private benefit under stressful conditions, this private benefit does not outweigh the cost **under our examined experimental conditions.**"

356: Good cautious phrasing.

No action required